# Exploration of nuclear body-enhanced sumoylation reveals that PML represses 2-cell features of embryonic stem cells

Sarah Tessier [1,2,7,10], Omar Ferhi[1,2,10], Marie-Claude Geoffroy [1,2,10], Román González-Prieto [3,8], Antoine Canat [2,9], Samuel Quentin[2,4], Marika Pla[5], Michiko Niwa-Kawakita [2], Pierre Bercier[1,2], Domitille Rérolle[1,2], Marilyn Tirard[6], Pierre Therizols [2], Emmanuelle Fabre [2], Alfred C. O. Vertegaal[3], Hugues de Thé [1,2,4] ✉ & Valérie Lallemand-Breitenbach [1,2] ✉

Membrane-less organelles are condensates formed by phase separation whose functions often remain enigmatic. Upon oxidative stress, PML scaffolds Nuclear Bodies (NBs) to regulate senescence or metabolic adaptation. PML NBs recruit many partner proteins, but the actual biochemical mechanism underlying their pleiotropic functions remains elusive. Similarly, PML role in embryonic stem cell (ESC) and retro-element biology is unsettled. Here we demonstrate that PML is essential for oxidative stress-driven partner SUMO2/3 conjugation in mouse ESCs (mESCs) or leukemia, a process often followed by their poly-ubiquitination and degradation. Functionally, PML is required for stress responses in mESCs. Differential proteomics unravel the KAP1 complex as a PML NB-dependent SUMO2-target in arsenic-treated APL mice or mESCs. PML-driven KAP1 sumoylation enables activation of this key epigenetic repressor implicated in retro-element silencing. Accordingly, *Pml*$^{-/-}$ mESCs re-express transposable elements and display 2-Cell-Like features, the latter enforced by PML-controlled SUMO2-conjugation of DPPA2. Thus, PML orchestrates mESC state by coordinating SUMO2-conjugation of different transcriptional regulators, raising new hypotheses about PML roles in cancer.

Membrane-less organelles are stress-sensitive deposits of concentrated bio-molecules. One specific type is nuclear bodies (NBs) scaffolded by the promyelocytic leukemia protein (PML), which attracts many disparate partner proteins. PML is required for essential processes such as senescence, metabolism, or viral restriction[1,2]. Restoration of PML NBs with arsenic or retinoic acid therapies underlies cures of patients with acute promyelocytic leukemia (APL), emphasizing the physio-pathological relevance of these compartments[3]. In vivo, PML senses reactive oxygen species (ROS), driving PML NB assembly and physiological responses to oxidative stress[4,5]. Arsenic directly binds PML, promoting its multimerization and NB assembly[6-9]. PML is also essential for the fitness of normal or malignant stem cells[10-13] and its down-regulation alleviates reprograming of mouse embryonic fibroblast (MEFs) into iPSCs[14]. These findings raise the question of how PML NBs contribute to stem cell biology and their response to stress.

Most PML NB partners recruited by the sumoylated PML scaffold were serendipitously discovered. PML shell insolubility has thwarted biochemical purification efforts to establish a comprehensive list of these partners. No generic biochemical activity has been demonstrated so far that might explain the diversity of PML-regulated physiological processes[1,2]. PML NBs have a proposed role in post-translational modifications of partners, including sumoylation, since UBC9, the SUMO-E2 enzyme, concentrates within NBs[1,2,9]. Functionally,

multi- or poly- conjugation by SUMO2/3 (two indistinguishable SUMO paralogues) may control proteasomal degradation through recruitment of SUMO-targeted ubiquitin ligases[7,15–18]. Sumoylation may also regulate transcription[19] and sustain epigenetic modifications, notably those essential for maintaining the identity of mouse embryonic stem cells (mESCs)[20–22]. In various cell lines, oxidative stress increases sumoylation, while paradoxically SUMO enzymes are inhibited by ROS[23–26]. These observations raise the question of whether sumoylation control might be involved in some PML-mediated functions, in particular in stem cell fate.

Here we establish that arsenic-enhanced PML NB biogenesis drives sumoylation and we identify an unbiased list of protein targets from primary cells. By establishing a biochemical function for PML NBs, our results explain the pleiotropy of PML physiological roles. Remarkably, in mESCs, PML favors SUMO2 conjugation of the KAP1 epigenetic regulatory complex, as well as that of the master transcription factor DPPA2 to oppose 2-Cell-Like (2CL) features, unraveling an unexpected key role for PML in the homeostasis of pluripotent ESCs.

## Results

### PML allows stress-induced mESC growth inhibition and SUMO2 conjugation in vivo

In pluripotent mESCs, PML expression is high and NBs are abundant, compared to early stages of differentiation induced by retinoic acid treatment and LIF withdraw (Fig. 1a). Arsenic stress rapidly increased PML NB assembly and recruited SUMO1/2/3 (Supplementary Fig. 1a), as in transformed cell lines[6]. To functionally explore any role of PML NBs in mESC homeostasis and stress response, we generated CRISPR/Cas9-engineered Pml knock out ($Pml^{-/-}$) mESCs. Critically, in two independent clones, PML was required for full arsenic-driven growth arrest (Fig. 1b), implying that the dynamics of NB assembly plays a key role in mESCs stress response.

Similar to PML expression, global SUMO2 conjugation was higher in undifferentiated mESCs than in their committed counterparts (Fig. 1a, right). To explore any role of PML NBs in sumoylation control, we leveraged mESCs and two in vivo biological systems wherein NB biogenesis is tunable by arsenic treatment (Fig. 1c). First, in $Pml^{+/+}$ and $Pml^{-/-}$ mESCs, we stably expressed $His_{10}$-SUMO1 or 2 (ref. 27) at low levels compared to endogenous SUMO peptides (Fig. 1d-inputs). Second, we expressed low $His_{10}$-SUMO2 level in an APL mouse model, where PML is expressed, but NB formation is impaired, a phenotype rapidly reversed by arsenic[3,6] (Fig. 1c, Supplementary Fig. 1b). Third, using $His_6$-HA-tagged Sumo1 knock-in mice[28], we derived $His_6$-HA-Sumo1;Pml^{+/+}$ and $His_6$-HA-Sumo1;Pml^{-/-}$ mice (Fig. 1c). In this setting, PML and SUMO expressions can first be boosted by polyI:C (pI:C)[29,30] (Supplementary Fig. 1c) and then PML NBs biogenesis can be enhanced by arsenic (Supplementary Fig. 1d).

To assess basal and arsenic-induced sumoylation, we first performed His-pulldown enrichment and Western blot analyses. In mESCs, arsenic rapidly promoted conjugation by SUMO1 and SUMO2/3 only in $Pml^{+/+}$ cells (Fig. 1d left and right respectively, quantification Supplementary Fig. 1e). Moreover, in untreated $Pml^{-/-}$ mESCs, basal $His_{10}$-SUMO2 conjugation was less efficient than in their $Pml^{+/+}$ counterparts (Fig. 1d, right arrowed). In contrast, basal SUMO1 conjugation was reproducibly slightly higher in $Pml^{-/-}$ mESCs, suggesting that SUMO1 might compensate for the SUMO2/3 conjugation defect (Fig. 1d, left panel arrowed). Similarly, in $His_{10}$-SUMO2-APLs, we observed a drastic global SUMO2 hyper-conjugation upon NB restoration by arsenic treatment (Fig. 1e, and Supplementary Fig. 1b, f). Finally, in the liver of $His_6$-HA-Sumo1 knock-in mice, arsenic treatment increased $His_6$-HA-SUMO1 conjugation, but also massively induced SUMO2/3-modification of SUMO1 conjugates, an effect sharply enhanced by pI:C priming (Fig. 1f, quantification Supplementary Fig. 1g). Here again, arsenic-enhanced sumoylation was not observed in the $Pml^{-/-}$ background. Importantly, experiments in $HA$-$His_{10}$-Sumo3 knock-in mice (encoding

the SUMO2 paralogue) demonstrated that arsenic-enhancement of direct SUMO2 conjugation in liver was of much greater amplitude than the one detected by SUMO1 pull-down (Fig. 1f, g). Collectively, all three models converge to demonstrate that stress-induced PML NB assembly correlates with global hyper-SUMO2 conjugation. In mESCs, such PML/SUMO2 coupling was even observed in the basal unstressed state.

SUMO2/3 being prone to its own sumoylation[20], we questioned PML contribution to the formation of SUMO2-containing chains. In both mESCs and APL mice, stable expression of a $His_{10}$-SUMO2 mutant defective for chain formation (SUMO2K/R)[27] blocked arsenic-increased global SUMO2 conjugation (Fig. 1e and Supplementary Fig. 1h), while critically, arsenic-induced PML multi-sumoylation was preserved. In untreated APL mice, the level of basal global conjugates was sharply increased in the $His_{10}$-SUMO2K/R pulldown (Fig. 1e, right part), suggesting that this mutant blocks basal SUMO2 chain formation and the subsequent degradation of targets, leading to their accumulation.

To mechanistically explore the contribution of NBs to PML- and arsenic-controlled sumoylation, we investigated whether the latter depends on partner protein recruitment, which requires a specific sumoylation site in PML[7,9]. We thus generated a E167R Pml knock-in mouse mutated on the corresponding sumoylation consensus sequence and derived $Pml^{E167R/-}$; $His_6$-HA-Sumo1 mice. As expected, PML NB-localization of endogenous DAXX, a well-known PML partner, was impaired in these mice (Supplementary Fig. 1i). Critically, global hyper-sumoylation by SUMO1 and SUMO2/3 upon arsenic/pI:C co-treatment was abolished in liver tissues of $Pml^{E167R/-}$; $His_6$-HA-Sumo1 mice (Fig. 1h right part, control PML profile in Supplementary Fig. 1j). Collectively, these findings support the conclusion that PML NBs, through partner recruitment, favor their sumoylation, primarily by SUMO2/3 chain formation.

### PML NBs favor stress-induced SUMO2-dependent ubiquitination

Given that hyper-sumoylation may drive poly-ubiquitination and degradation, we then assessed the fate of PML-driven sumoylated proteins upon arsenic stress. We found that sumoylated proteins decreased rapidly after their initial boost in $Pml^{+/+}$ mESCs (Fig. 2a and Supplementary Fig. 2a) or liver (Fig. 2b). Critically, in both models, this decrease was accompanied by a PML-dependent wave of dual SUMO2-specific/ubiquitin conjugation (Fig. 2a, b and Supplementary Fig. 2b) and accumulation of poly-ubiquitinated proteins in PML NBs (Supplementary Fig. 2c). In vivo pre-treatment with Bortezomib, a proteasome inhibitor, stabilized SUMO2/3 conjugates upon arsenic exposure in Pml-proficient livers or APL (Supplementary Fig. 2d and Supplementary Fig. 4b below). In the experiments above, we cannot exclude that a fraction of SUMO2-conjugates consists of sumoylated PML. Note, however, that the kinetic changes in PML profiles upon arsenic treatment do not match that of SUMO2 (Fig. 2b) and that SUMO2K/R altered the basal and arsenic-induced SUMO2 conjugates, but not those of PML (Fig. 1e). Collectively, these observations support the idea that PML NBs enables SUMO2-dependent ubiquitination and subsequent degradation of a substantial amount of targets under oxidative stress ex vivo or in vivo.

Arsenic-driven PML NB formation may concentrate the SUMO-E2 UBC9 within NBs (Supplementary Fig. 2e)[9]. In cell lines and in vitro, SUMO and ubiquitin modifying enzymes may be inactivated by ROS, which oxidizes their active-site cysteines[24,26]. Since PML is also a cysteine-rich oxidation-prone protein[8], we wondered whether PML shell might constitute a local oxidation-protective environment. We thus explored the redox status within PML NBs using the glutathione Grx1-roGFP2 sensor[31] in fusion with the NB-associated SP100 protein (Supplementary Fig. 2f). Grx1-roGFP2-SP100 was localized at PML NBs, as expected, and was significantly less oxidized than in the rest of the nucleoplasm. Thus, lower oxidation levels at PML NBs suggests that PML NBs could shield sumoylation enzymes such as UBC9 from inhibition by cellular ROS.

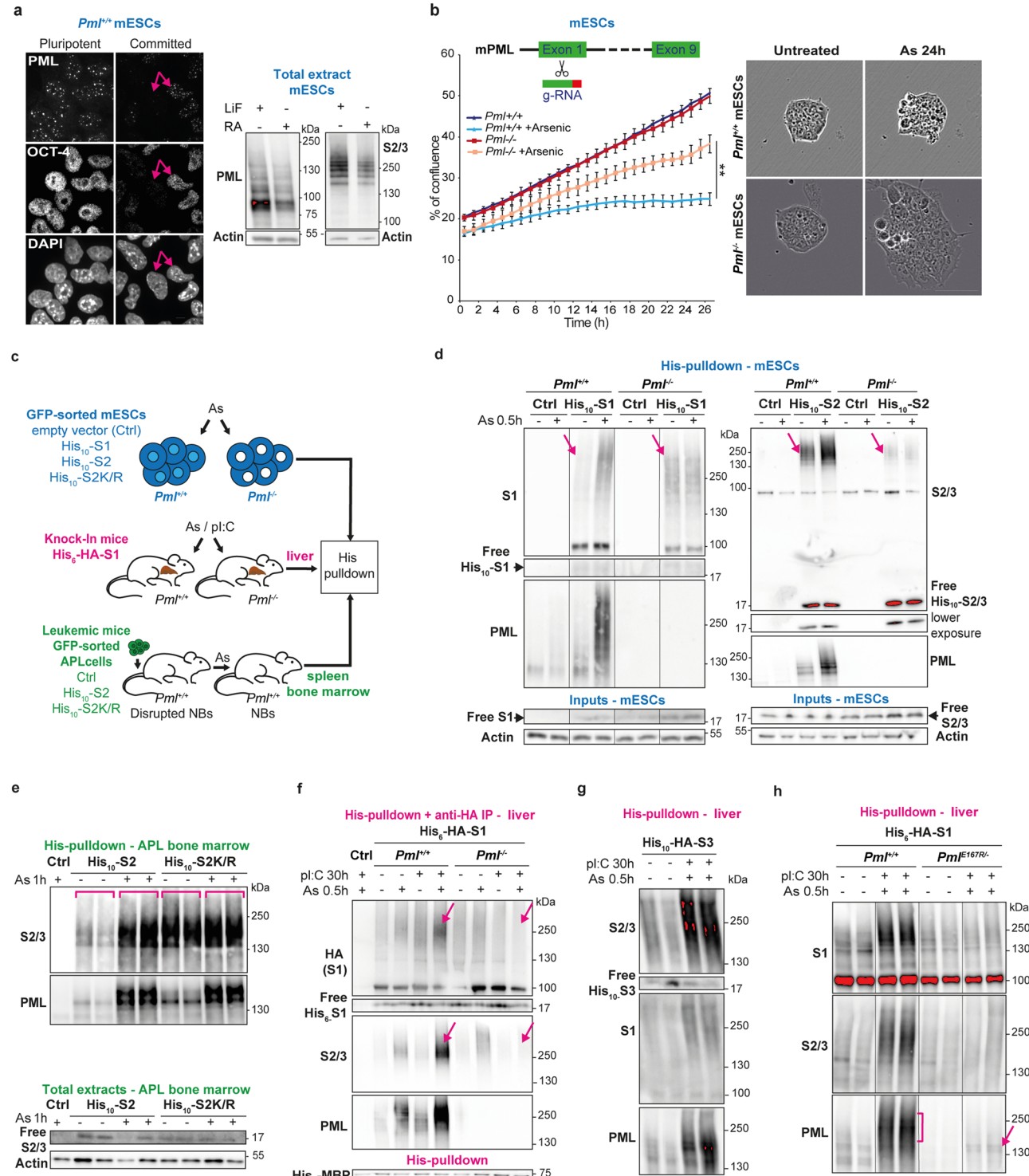

## Identification of PML NB-dependent SUMO targets

To identify the SUMO conjugates regulated upon PML NB assembly, we undertook large-scale purifications of sumoylated proteins in APL mice and in mESCs for label-free quantitative proteomic analysis (LFQ LC-MS/MS)[27] (Fig. 3a). The spleen is a hematopoietic organ enlarged upon leukemic cell invasion, rendering possible in vivo differential analysis of SUMO2 targets with altered (untreated APL) or restored (arsenic-treated APL) PML NBs formation. Three cohorts of His[10]-SUMO2- expressing or control APL mice were injected or not with arsenic and sampled after 1 h (cohort no. 1), 3 h (cohort no. 2) or 6 h (cohort no. 3). To identify proteins conjugated specifically by His[10]-SUMO2, we compared, within

each cohort, His[10]-SUMO2 APL versus control APL, in both treated and untreated animals (see methods and Supplementary Data 1). We thus identified reliable His[10]-SUMO2-conjugated proteins, with over 60% overlap in untreated animals between the three cohorts (Supplementary Data 1). In untreated mice, PML and PML/RARA were the top SUMO conjugates, reflecting their known efficient baseline conjugation. Most of these in vivo identified basal SUMO2 targets belonged to canonical SUMO-regulated pathways and key hematopoietic regulators, notably C/EBPα[32] (Supplementary Data 1).

We then focused on targets differentially conjugated by SUMO2 upon arsenic-driven NB reassembly (Fig. 3b). To do so, we

**Fig. 1 | PML is required for stress response and sumoylation in mESCs and mice. a** PML NBs (representative confocal analysis, left) and PML expression (Western blot analysis, right) in pluripotent (LIF) versus committed (Retinoic Acid and LIF withdrawal) mESCs, in which Oct4 decreases (compare arrowed cell). Scale: 5 μm. Representative data from three independent experiments. **b** IncuCyte cell proliferation assay showing arsenic stress-resistance with loss of Pml; top schematic of Pml edition. Mean +/− SD of $n = 4$ independent biological replicates, right bar indicates adjusted p-value, **$p = 0.007$, between growth curves of arsenic-treated $Pml^{+/+}$ and arsenic-treated $Pml^{-/-}$ mESCs, one-way ANOVA followed by Sidak's multiple comparisons test (left); and representative images (right), Scale: 50 μm. Representative of 2 CrispR/Cas9-generated $Pml^{-/-}$ mESC clones. **c** Experimental mouse models with tunable PML NBs: (Blue) $Pml^{+/+}$ or $Pml^{-/-}$ mESCs stably expressing His10-SUMO1-, His10-SUMO2-, His10-SUMO2K/R-IRES-GFP or GFP (His10-S1, His10-S2, His10-S2K/R, Ctrl). (Pink) His6-HA-Sumo1 knock-in; $Pml^{+/+}$ or $Pml^{-/-}$ mice injected with arsenic and pI:C to maximize NBs. (Green) APL mice obtained from serial transplantations of APL cells (Ctrl APL) or sorted APL cells expressing the indicated His10-SUMOs, with arsenic injection to restore PML NBs. **d** Pulldown (PD)

of His10-S1 (left) or His10-S2 (right) conjugates showing arsenic-increased sumoylation in $Pml^{+/+}$ mESCs only. Inputs indicating similar low levels of free His10-SUMO expression. Arrows: baseline sumoylation depending on PML. Representative data. Supplementary Fig. 1e for statistics. **e** PD from APL bone marrows as in (1d), showing increases in His10-S2, but not His10-S2K/R, conjugation upon arsenic injection. Total extracts indicate respective levels of free S2/3. Representative experiment (2 mice per condition, 1 Ctrl APL with untagged SUMO2), $n = 3$. **f** His6-HA-S1 conjugates dually purified from liver, showing PML-dependent increase in S1/2/3 conjugation upon arsenic/pI:C (pink arrows). Ctrl: $Pml^{+/+}$ mice, His6-MBP recombinant protein: internal PD control. See Supplementary Fig. 1g for statistics. **g** Same as (1f), using $His_{10}$-HA-Sumo3 knock-in mice. **h** Arsenic-increased sumoylation is lost in $Pml^{E167R/-}$ knock in mice with PML NBs defective for partner recruitment, PD from the indicated mice, as in (1f). Control with PML profile from total extract in Supplementary Fig. 1j. **g, h** Representative of $n = 6$ mice per condition over three independent experiments. Source data are provided as a Source Data file.

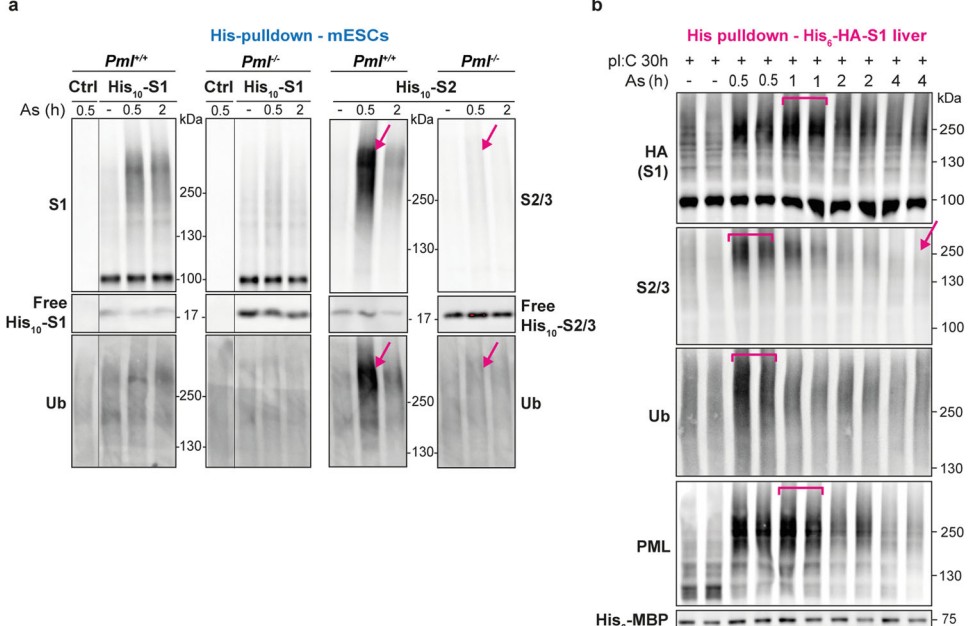

**Fig. 2 | PML NBs are required for SUMO2/ubiquitin conjugation in vivo. a** His-PD of His10-S1 or His10-S2 conjugates from mESCs exposed to arsenic, revealing a rapid ubiquitination of the SUMO2 conjugates only, after their PML-dependent boost. The levels of the various free His10-SUMOs in mESCs are indicated. $n = 5$

independent experiments. **b** Representative of His pulldown from His6-HA-SUMO1 mouse liver showing waves of SUMO1 and multi-SUMO1, SUMO2/3 or Ubiquitin (Ub) conjugation with time of arsenic injection. 2 mice per condition are shown over 2 independent experiments with 3 mice per condition.

selected proteins significantly increased or decreased in arsenic-treated His10-SUMO2 APL compared to untreated His10-SUMO2 APL mice, within each cohort. The majority of the 88 arsenic-modulated targets identified among the 3 cohorts had increased sumoylation levels (Fig. 3b and Supplementary Data 2). Yet the abundance of those targets in the total APL proteome was unchanged upon arsenic (Supplementary Fig. 3a and Supplementary Data 2, 3), implying that arsenic actually enhanced their sumoylation, rather than their abundance. As expected, these 88 arsenic-modulated SUMO2 targets were enriched in known PML-interacting proteins (Fig. 3c)[33]. One hour after injection, when NBs started to re-organize, 19 targets displayed an increased sumoylation (Fig. 3b left, Supplementary Data 1–2), including PML/RARA itself, negative regulators of transcription and epigenetic regulators, some of which may bind RARA (NCOR1 or EPC2, the polycomb EZH2 partner) (Fig. 3d). Strikingly, among these early arsenic-modulated SUMO2 conjugates, we identified five proteins belonging to the same chromatin-remodeling complex involved in stem cell maintenance,

comprising KAP1 (TRIM28 or TIF1beta), CHAF1a subunit of the CAF-1 histone chaperone, the MMSET histone methyl transferase (also known as WHSC1 or NSD2) and two DNA-docking KRAB-Znf proteins (ZNF710, ZNF148) (Fig. 3b, e and Supplementary Data 2)[34,35]. Other members of the KAP1 complex, not identified in this experiment, are known PML NBs partners, such as the DAXX/ATRX H3.3 histone chaperone or the heterochromatin protein 1 (HP1) (Fig. 3e)[35–37].

Remarkably, the repertoire of arsenic-modulated His10-SUMO2 targets underwent significant changes with time (Fig. 3b and Supplementary Data 2). Sumoylation of proteins linked to stem cell potential, such as PML/RARA, KAP1 or Polycomb complex member (PHC2), all decreased at later timepoints, likely reflecting their SUMO2-triggered degradation (Fig. 3b and Supplementary Data 1–2). We also identified increases in sumoylation of SP100, TDG, MORC3 and NFIL3, known NB components, at later time points (Supplementary Fig. 3b)[9]. Several SUMO2 targets identified here -such as CCAR1, MORC3, KAP1- may contribute to the poorly understood PML control over p53 function (Fig. 3b and Supplementary Data 2)[3,36,38,39].

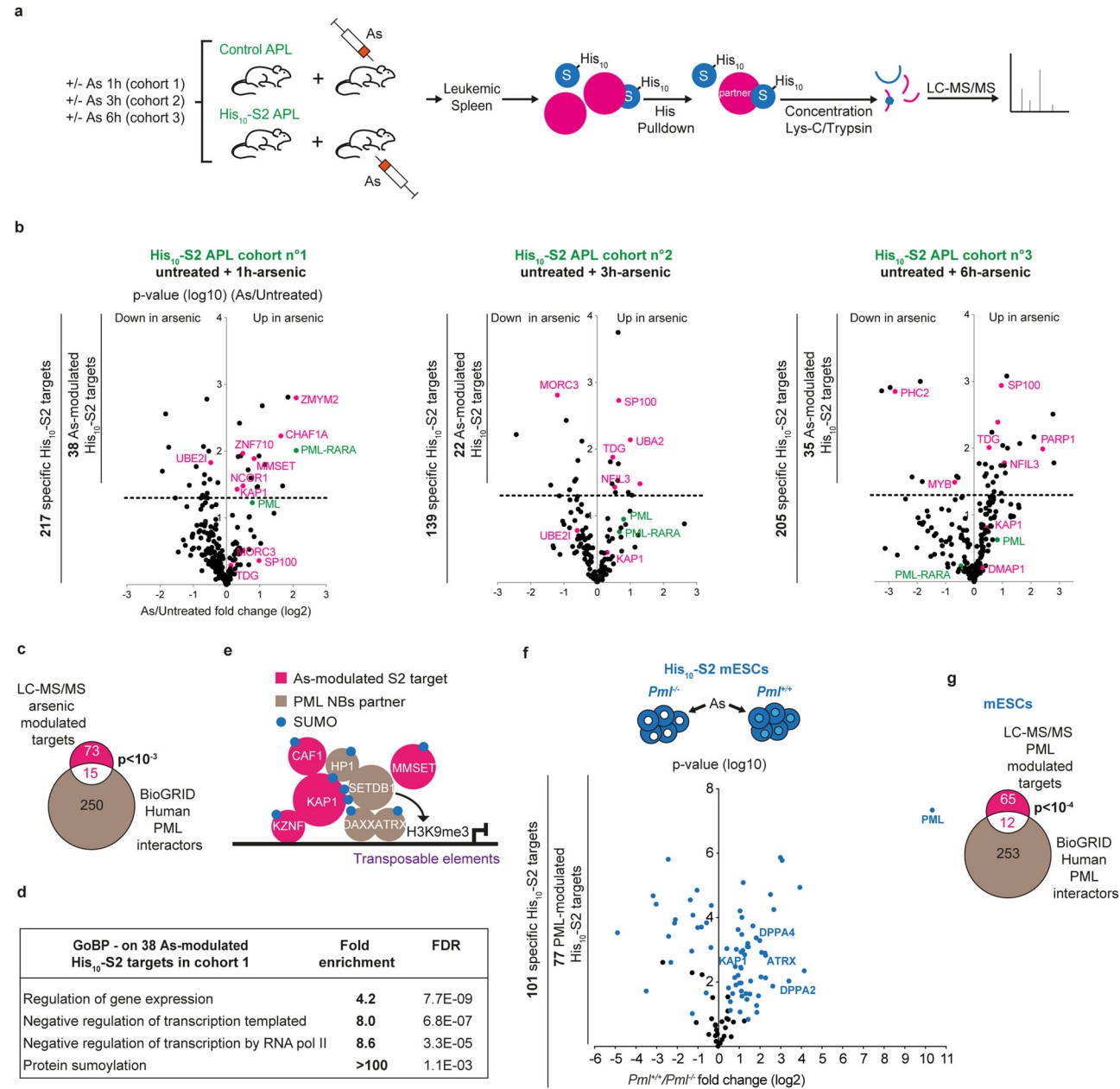

**Fig. 3 | Identification of the His$_{10}$-SUMO2 targets upon arsenic-induced PML NB reorganization in APL mice. a** Experimental setup for differential SUMO-proteome analyses upon arsenic, based on large-scale PD of His$_{10}$-S2 conjugates from three mouse cohorts, each containing 6 Ctrl and 6 His$_{10}$-S2 APL mice treated ($n = 3$) or not with arsenic ($n = 3$), for the indicated time, followed by LC-MS/MS analyses. **b** Volcano plots showing proteins differentially sumoylated in arsenic- vs untreated His$_{10}$-S2 APL mice for each cohort; label-free quantification (LFQ) fold changes; $p$-values, two-sided $t$-test; dashed line represents p-value threshold (0.05). Targets plotted when significantly enriched in His$_{10}$-S2 vs Ctrl APL mice (Supplementary Data 1), numbers of identified proteins in each group are indicated (left bars). **c** Venn diagram showing proteins common to arsenic-modulated His$_{10}$-S2 targets identified in (**b**) and previously reported PML partners in ref. 33 and

Biogrids; $p$-value (proportion of PML interactors in arsenic-modulated His$_{10}$-S2 targets), two-sided standard $t$-test. **d** Gene Ontology enrichment analysis (GOBP, gene ontology biological process) of 1h-arsenic-modulated targets, highlighting repressive transcriptional regulation. **e** Schematic overview of the arsenic-modulated SUMO2 targets (pink) and other members related to PML in the KAP1 complex. **f** Volcano plot showing His$_{10}$-SUMO-2 conjugated proteins (blue) identified by LC-MS/MC analysis in His$_{10}$-S2 *Pml$^{-/-}$* vs *Pml$^{+/+}$* mESCs (see supplementary Fig 3c for Western blot controls). $n = 4$ biological replicates; fold changes (log2), $p$-value (log10), two-sided $t$-test. **g** Venn diagram showing proteins common to PML-dependent His$_{10}$-S2 targets identified in (**f**) and reported as PML partners as in (**c**), $p$-value two-sided $t$-test.

To further support the relevance of our APL model for the identification of PML NB-regulated proteins, we performed the same SUMO proteomic analyses using mice engrafted with *Pml*-deficient APL cells (Supplementary Data 4). We compared *Pml$^{-/-}$* APL cells expressing or not His$_{10}$-SUMO2, from leukemic mice treated or not with arsenic for 3 h. Among the 554 His$_{10}$-SUMO2 targets in *Pml$^{-/-}$* APL mice, 86 were differentially conjugated in arsenic-treated mice. Remarkably, only a

few arsenic-modulated His$_{10}$-SUMO2 targets identified in the previous *Pml$^{+/+}$* APL cohorts also varied upon arsenic treatment in *Pml*-deficient APL mice, PML/RARA being the top enhanced one (Supplementary Data 4). In particular, in absence of *Pml*, none of the previously identified members of KAP1 complex and other PML partners (TDG, MORC3, SP100) underwent arsenic-modulated change in sumoylation, consistently with their PML NBs-regulated SUMO2 conjugation.

Overall, these unbiased explorations of stress-modulated SUMO2-conjugation in vivo establish the sumoylation activity of PML NBs and identify a critical epigenetic regulatory complex as a key target.

We also analyzed the His$_{10}$-SUMO2 targets differentially conjugated in $Pml^{+/+}$ and $Pml^{-/-}$ mESCs (Supplementary Fig. 3c), treated with arsenic to maximize PML NB assembly. Strikingly, among the 101 proteins identified as specific His$_{10}$-SUMO2 targets (by comparison with control mESCs that do not express tagged SUMO2), 77 proteins were differentially sumoylated in $Pml^{+/+}$ vs $Pml^{-/-}$ mESCs, with again a significant enrichment in known PML-interacting proteins (Fig. 3f, g and Supplementary Data 5). Importantly, 54 among these 77 His$_{10}$-SUMO2 targets were increased in $Pml^{+/+}$ compared to $Pml^{-/-}$ mESCs, among which were KAP1 and its interacting NBs-associated partner, the ATRX H3.3 chaperone[40] (Fig. 3e, f). His$_{10}$-SUMO2 conjugation of components of the polycomb repressive complexes (L3mbtl2, MGA, SUZ12) was increased in $Pml^{+/+}$ mESCs (Supplementary Data 5). Interestingly, we also identified as PML-dependent SUMO2 targets, DPPA2 and DPPA4, members of a transcription factor complex controlling mESCs identity (see below). Altogether, differential SUMO2 proteomic profiling identified various targets of PML NBs in treated APL and mESCs, not only KAP1 and its interacting chromatin-remodeling partners, but also key transcription regulators, suggesting a broad role of PML in epigenetic and transcriptional regulation.

## PML favors KAP1 sumoylation and TE repression

Since sumoylation is a critical activator of KAP1 complex function[34,41] (Fig. 3e), we explored the role of PML and SUMOs in KAP1-dependent epigenetic repression in mESCs. We first established that multi-SUMO2 KAP1-conjugates were enriched in $Pml^{+/+}$ mESCs relative to $Pml^{-/-}$ mESCs in basal conditions (Supplementary Fig. 4a). Then, arsenic sharply increased, in a PML-dependent manner, KAP1 conjugation by His$_{10}$-SUMO2, but not by His$_{10}$-SUMO1 (Fig. 4a and Supplementary Fig. 4a). In APL mice pre-treated with a proteasome inhibitor, KAP1 was also hyper-conjugated by SUMO2 upon arsenic treatment (Fig. 4b, controls in Supplementary Fig. 4b). Finally, KAP1-YFP had a nuclear diffuse localization, but it overlapped with PML NBs upon proteasome inhibition (Fig. 4c)[42]. Collectively, KAP1 behaves as a PML partner undergoing stress-enhanced SUMO2 conjugation, a key functional modification for this master epigenetic regulator.

KAP1 multi-sumoylation drives its interactions with HP1 and the SETDB1 methyltransferase, allowing trimethylation of lysine 9 of histone H3 and transcription repression[34,35,41] (Fig. 3e). In mESCs, the KAP1 complex represses endogenous retroviruses (ERVs)[40,43], and the SUMO pathway is essential for this silencing[22,34]. Using chromatin immunoprecipitation, we first observed that H3K9me3 was significantly less abundant at the 5′ LTRs of MLV, IAPEz, EtnERV2, and MusD elements of the ERV1 and ERVK families in $Pml^{-/-}$ versus $Pml^{+/+}$ mESCs (Fig. 4d). In contrast, PML-deficiency had little effect on H3K9me3 modification at the LINE-1 promoter or the MERVL LTRs (Fig. 4d), consistent with their indirect repression by KAP1 reported in refs. 44,45. Accordingly, ERV and LINE-1 transcripts were derepressed in $Pml^{-/-}$ mESCs, as indicated by RT-qPCR quantification (Fig. 4e). Then, in RNA-sequencing (RNA-seq) analysis for Transposable Element (TE) multi-mapped reads[46], we found 73 repeated elements miss-regulated more than 4-fold upon PML deficiency, 68 of which were upregulated (Fig. 4f and Supplementary Data 6), primarily KAP1-repressed ERVs and LINE-1. Altogether, these observations imply that PML contributes to silence TEs in mESCs, most likely through its regulation of KAP1 sumoylation and repressive activity. Supporting this view, MORC3, another PML-dependent SUMO2 target and KAP1 partner (Fig. 3e and Supplementary Data 5), is involved in endogenous retroviral epigenetic silencing[47].

To strengthen the links between PML-driven KAP1-sumoylation and ERV repression, we first confirmed that down-regulation of UBC9

or KAP1 increased the expression of ERVK (Etn-ERV2 and IAPz)[40,48], similar to $Pml$ knockout, although to a lesser extend (Fig. 4g and Supplementary Fig. 4c). To test whether the KAP1 complex is inactive in absence of PML due to defects in its sumoylation, we compared the effect of $Kap1$ siRNA on ERVs in $Pml^{+/+}$ and $Pml^{-/-}$ mESCs. While $Kap1$ knockdown derepressed ERVs in $Pml^{+/+}$ mESCs, $Kap1$ siRNA had no significant effect in $Pml^{-/-}$ mESCs (Fig. 4h). Similarly, sumoylation inhibition by the ML792 SUMO-activating enzyme poison increased ERVK in wildtype mESCs, while it had no effect in $Pml^{-/-}$ mESCs (Fig. 4i and Supplementary Fig. 4d). Finally, KAP1-YFP transduction in $Pml^{-/-}$ mESCs rescued ERVK repression (Fig. 4j and Supplementary Fig. 4f). This rescue was impaired when sumoylation was inhibited by ML792. Collectively these findings support the model that PML maintains ERV repression by controlling KAP1 sumoylation in mESCs.

## PML opposes 2-cell like features of mESCS

Many TEs are transiently expressed in totipotent 2-cell (2 C) embryos, but become repressed in pluripotent blastocysts (from which ESCs are derived). A fraction of mESCs oscilates in and out of a 2-Cell-Like (2CL) state, recapitulating some aspects of zygotic genome activation (ZGA), a process inhibited by the KAP1 complex. MERVL expression is a hallmark of 2CL transition[49–51] and was deregulated in $Pml^{-/-}$ mESCs (Fig. 4e and Supplementary Data 6). To investigate whether the loss of PML could also affect pluripotent to 2CL transition of mESCs, we compared transcriptomes of $Pml^{-/-}$ and $Pml^{+/+}$ mESCs. Indeed, we found highly significant differential enrichment in the gene set encompassing both 2C- and 2CL-restricted transcripts[49] (Fig. 5a). The top upregulated genes in $Pml^{-/-}$ mESCs include key 2CL markers such $Zscan4$, $EiF1a$ and $EiF1a$-like gene families ($Zscan4b$-e, Gm2016, Gm5662, Gm4340), as well as $Piwil2$ (Fig. 5b, Supplementary Fig. 4g and Supplementary Data 6). Expression of the $Dux$ master activator, another hallmark of 2 C embryos or 2CL mESCs[50,51], was also upregulated in the absence of $Pml$ (Fig. 5c). In contrast, transcripts of pluripotent markers, Nanog, Kfl4 or Oct4 did not significantly change with $Pml$ loss in our mESCs pool (Figs. 4f, 5g and Supplementary Data 6). Finally, $Pml^{-/-}$ mESCs exhibited downregulation of MYC and oxidative phosphorylation pathways (Fig. 5d, Supplementary Fig. 4e and Supplementary Data 6), another feature of 2CL cells[52,53]. Collectively, PML appears to oppose the transcriptional 2CL program.

In addition to the KAP1 repressor, DPPA2 is a master positive transcriptional regulator of ZGA, upstream of $Dux$, both being required for the establishment of 2CL state[45,50,54]. Remarkably, the expression of $Mael$ and $Prex2$, two Dux-independent targets of DPPA2[45], were upregulated in $Pml^{-/-}$ mESCs (Supplementary Data 6 and Fig. 5c). The transcriptional activity of DPPA2 and its role to promote 2CL conversion are blocked by its sumoylation[54,55]. Critically, our mESC SUMO proteomic analysis identified both DPPA2 and DPPA4 as PML-dependent SUMO2 targets (Fig. 3f and Supplementary Data 5). We confirmed basal and arsenic-enhanced DPPA2 conjugation by His$_{10}$-SUMO2 pulldown in $Pml^{+/+}$, but not in $Pml^{-/-}$ mESCs (Fig. 5e, controls in Supplementary Fig. 4h). In contrast, His$_{10}$-SUMO1 conjugation of DPPA2 was low, independent of $Pml$, and not modulated by arsenic, similar to that of KAP1 (Fig. 4a). Sumoylation inhibition by ML792 increased the DPPA2-target $Dux$ and $Mael$ transcripts in $Pml^{+/+}$ mESCs, while it had weaker or no effects in $Pml^{-/-}$ mESCs, wich have no basal DPPA2 sumoylation (Supplementary Fig. 4i). Thus, in addition to its role on KAP1 activation, PML may also maintain silencing of 2CL genes by controlling DPPA2 repressive sumoylation. Other PML-regulated SUMO2 conjugates identified by the mass spectrometry analyses, such as CHAF1a, ZMYM2, as well as the KDM5B H3K4me2/3 demethylase (Fig. 3e and Supplementary Data 2, 5), may also oppose 2CL transition[51,56,57]. To confirm that PML plays a role in the 2CL transition (in addition to mere proteomic/transcriptomic signatures), we assessed the number of mESCs with a 2CL feature by quantifying

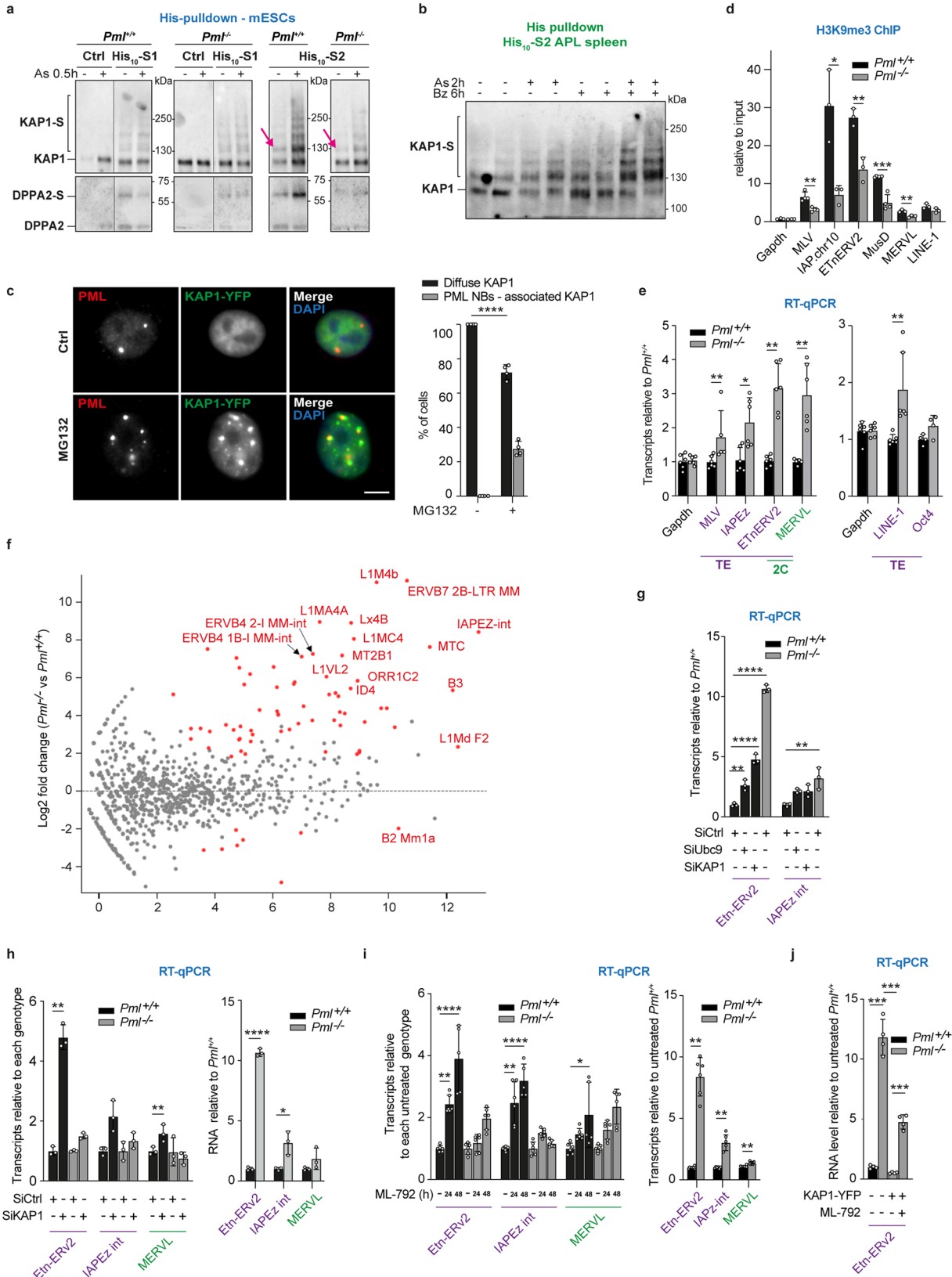

ZSCAN4-positive and OCT4-negative cells[58]. Remarkably, these cells were significantly enriched in absence of *Pml* (Fig. 5f).

Ground state ESCs is obtained by triggering global DNA demethylation in 2i culture media with vitamin C[59]. In this setting, loss of *Pml* led to an even more drastic derepression of 2CL genes (Fig. 5g). Interestingly, *Pml⁻/⁻ mESCs* stopped growing and became apoptotic when grown in 2i media, implying an essential role for PML in survival (Fig. 5h and Supplementary Fig. 4j). Collectively, these data establish a role for PML in restricting features of 2CL mESCs transition by enforcing SUMO2-conjugation of multiple positive (such as DPPA2) and negative (such as KAP1) master regulators.

**Fig. 4 | PML controls KAP1 sumoylation leading to TE de-repression in *Pml*[-/-] mESCs. a** PML-dependent arsenic-enhanced S2-conjugation of KAP1 and DPPA2 as in Fig. 2b (corresponding SUMO blots). Arrows indicate the baseline S2 adduct on KAP1. Representative data, $n = 5$ independent biological replicates. **b** Arsenic-increased sumoylation of KAP1 in His$_{10}$-S2 APL mice, stabilized by bortezomib pre-injection. PD $n = 2$ mice per condition; representative of two independent experiments (see Supplementary Fig. 4b for KAP1 inputs and S2/3 conjugates). **c** Representative images of KAP1-YFP (green) localization overlapping with PML NBs (red) upon MG132 in HeLa cells, scale: 5μm (left). Percentage of cells with KAP1-YFP colocalization with PML NBs, $n = 60$ (untreated), 100 (MG132-treated) cells, Mean +/− SD of $n = 4$ independent biological replicates, $****p < 0.0001$ Two-tailed unpaired Student's $t$-test (right). **d** ChIP analysis showing decrease of the H3K9me3 mark at the promoter regions (LTRs or 5') of the indicated TEs in *Pml*[-/-] mESCs. Mean +/− SD of $n = 4$ biological replicates, relative to inputs and normalized on actin gene, $*p < 0.05$, $**p < 0.01$ two-tailed unpaired Student's $t$-test. Representative of $n = 3$ independent experiment. **e** Fold increase in the indicated TE transcripts, Mean +/− SD of $n = 6$ biological replicates from three independent experiments (right), normalized on actin and relative to paired values in *Pml*[+/+] mESCs, $***p < 0.001$; $****p < 0.0001$, two-tailed unpaired Mann−Whitney $t$-test.

Representative of two CrispR/Cas9-generated *Pml*[-/-] mESC clones. **f** MA-Plot of the RNAseq multireads analysis comparing *Pml*[-/-] to *Pml*[+/+] mESCs, showing increased TE expression. Red dots highlight significant TEs with >4 fold difference (see also Supplementary Data 6). **g** Fold change of the indicated ERV transcripts normalized on GAPDH, relative to siRNA Ctrl (Scr)-transfected *Pml*[+/+] mESCs, Mean +/− SD of $n = 3$ biological replicates $*p < 0.05$; $**p < 0.01$; $****p < 0.0001$, one-way ANOVA followed by Dunnett's multiple comparisons test. **h** Fold change of the indicated ERV transcripts in siKAP1-transfected mESCs related to their Ctrl (siRNA Scr) mESCs (left), fold changes in *Pml*[-/-] related to *Pml*[+/+] mESCs are indicated (right), Mean +/− SD of $n = 3$ biological replicates, $*p < 0.05$, $**p < 0.01$, $****p < 0.0001$, two-tailed unpaired Welch's test. **i** Fold change of the indicated ERV transcripts in *Pml*[+/+] and *Pml*[-/-] mESCs treated or not with 1uM ML792 for 24 h and 48 h, Mean +/− SD of $n = 6$ biological replicates, one-way ANOVA followed by Dunnett's multiple comparisons test (left); fold changes in *Pml*[-/-] related to *Pml*[+/+] mESCs are indicated, two-tailed unpaired Mann−Whitney $t$-test, $**p < 0.01$ (right). **j** Fold increase of Etn-ERV2 in *Pml*[-/-] expressing KAP1-YFP and treated or not with ML792, normalized to GAPDH and are relative to untreated *Pml*[-/-]. Mean +/− SD, $*p < 0.05$, two-tailed unpaired Welch's test. **g**−**j** Representative of three independent SD of $n = 4$ biological replicates experiments. Source data are provided as a Source Data file.

## Discussion

The abundance and integrity of PML NBs are challenged in a variety of physio-pathological conditions, ranging from oxidative stress to cancer or viral infections. The actual biochemical role of PML NBs was debated due to their pleiotropic biological activities and to the functional diversity of the one-by-one identified partner proteins. The formation of PML NBs is regulated by both the abundance of PML (depending on stem cell status, interferon or P53 signaling) and its oxidative stress-responsive dynamics, as explored with arsenic. Here, we formally demonstrate that PML NBs drastically enhance partner SUMO2 conjugation upon stress, subsequently regulating their stability ex vivo or in vivo. When PML is abundant, NBs may even facilitate conjugation in baseline conditions, as demonstrated for DPPA2 or KAP1 in mESCs, controlling partner functions (Fig. 6). From yeast to human, sumoylation plays a critical role in adaptive stress responses[24,60]. Distinct from earlier studies[61,62], our differential SUMO-proteomic evaluates stress responses in vivo, and identifies novel PML NB targets. Mechanistically, our data suggest that ROS-triggered cysteine-rich PML shells[4,8] may protect UBC9 from oxidation, although experiments identifying the impact of PML shells upon UBC9 oxidation and function would be required for formal demonstration in cellulo. While PML is not a bona fide SUMO-E3 ligase, our results support a model in which liquid-like condensates immobilize active enzymes and their substrates, to facilitate post-translational modifications of low abundance regulatory proteins[63,64]. In particular, PML NBs increase UBC9 processivity, enabling stress-induced global poly-SUMO2 modifications and proteasome degradation of partners[16,17,65] (Figs. 1 and 2). Other NB-associated UBC9 activators (PIAS, ARF) may further enhance PML NBs activity[66–68]. Furthermore, by sequestering SUMO1 or 2 away from the rest of the nucleoplasm (Supplementary Figs. 1a, 2e−f), PML NBs may also decrease their availability (together with that of UBC9) for non-NB-associated substrates, contributing to change of the spectrum of SUMO targets. In arsenic-treated APL mice, kinetic changes in the spectrum of SUMO targets suggest that the localization of PML -initially in close proximity to chromatin and later in reorganized NBs - may determine partner selectivity. Interestingly, our study unraveled number of putative RARA partners (KAP1, Polycomb), which undergo PML-facilitated sumoylation upon arsenic, suggesting that they may also be associated with PML/RARA. These key repressor complexes could thus contribute to PML/RARA-mediated gene silencing. Given its multiple roles, the KAP1 complex could enforce leukemogenesis through PML/RARA-driven target gene silencing, but later contribute to therapy-induced PML/p53-mediated senescence[35,69]. Beyond APL, PML-facilitated KAP1 repressive functions may actually contribute to senescence, viral latency[70,71] or cancer stem cell

biology[13,72,73]. The proposed PML roles in the regulation of chromatin status at specific loci may also involve KAP1 and/or SUMO2/3 conjugation[74–76].

Sumoylation sustains epigenetic repression in mESCs, contributing to repression of both ERVs and 2CL program[22,34,77]. The PML-dependent control of TE expression unraveled here could explain basal activation of interferon signaling in multiple types of *Pml*[-/-] cells[78,79] (our unpublished data). Precisely ordering the events controlling 2CL transition is currently under active investigation, in particular regarding DUX/ZSCAN4 and MERLV contributions[49,51,56,80]. By coordinating the action of various repressors and activators through orchestration of their SUMO2 conjugation, PML NBs oppose mESC 2CL-like features and TE expression (Fig. 6). More broadly, enhancement of partner sumoylation is likely a common mechanism through which PML NBs exert their pleiotropic effects to orchestrates cell fate, adaptive responses or proteostasis.

## Methods
### Cell culture and transduction

Mouse ESCs E14tg2a were obtained from 129/Ola blastocyst mice (P. Navarro (Pasteur Institute, Paris, France) and cultured in DMEM (Gibco, #41966029) supplemented with 10% Fetal Calf Serum (FCS), 1% non-essential amino acid (Gibco, #11140050), 1% Glutamax (Gibco, #35050-038), 1% Penicillin/Streptomycin (Gibco, #10378016), 1000 units/ml of recombinant Leukemia Inhibitory Factor (LIF, Interchim, #8V6280) and 0.1% β-mercaptoethanol (Merck, #M6250) on gelatin-coated plates. Ground-state mESCs were grown in serum-free 2i medium (half NeurobasalGibco, #21103049) and DMEM/F-12 (Gibco, #11320033)) supplemented with 0.5% N2 (Gibco, #17504048) and 1% B27 supplement (Gibco, #17504044), 1% Glutamax (Gibco, #35050-038), 1% Penicillin/Streptomycin (Gibco, #10378016), 0.2 mg/ml BSA (Thermofisher, #AM2616), 0.01% Monothioglycerol (Sigma, #M6145), 10000 units/ml LIF (Merck, #8V6280), 0.1% inhibitor cocktail (Merck, #PD0325901 & CHIR99021) and 1% Vitamin C (Sigma, #A4403). MEF (our laboratory), HeLa (ATCC), and Plat-E (Lavau C, USA) cells were grown in DMEM GlutaMAX (Gibco) supplemented with 10% FCS. All cells lines used were negative for mycoplasma (tested by Eurofins MWG France once a month) and were manipulated separately to avoid any cross contamination during cell passages. *Pml*[-/-] cells were regularly checked for the absence of PML by IF or WB. No Karyotype was performed.

APL cells or mESCs were transduced using retroviruses produced by Plat-E packaging cells, after transfection with MSCV-IRES-GFP constructs expressing His$_{10}$-SUMO1 or 2 paralogs (Effecten reagent, Qiagen, #301425). GFP-positive cells with similar and weak fluorescent

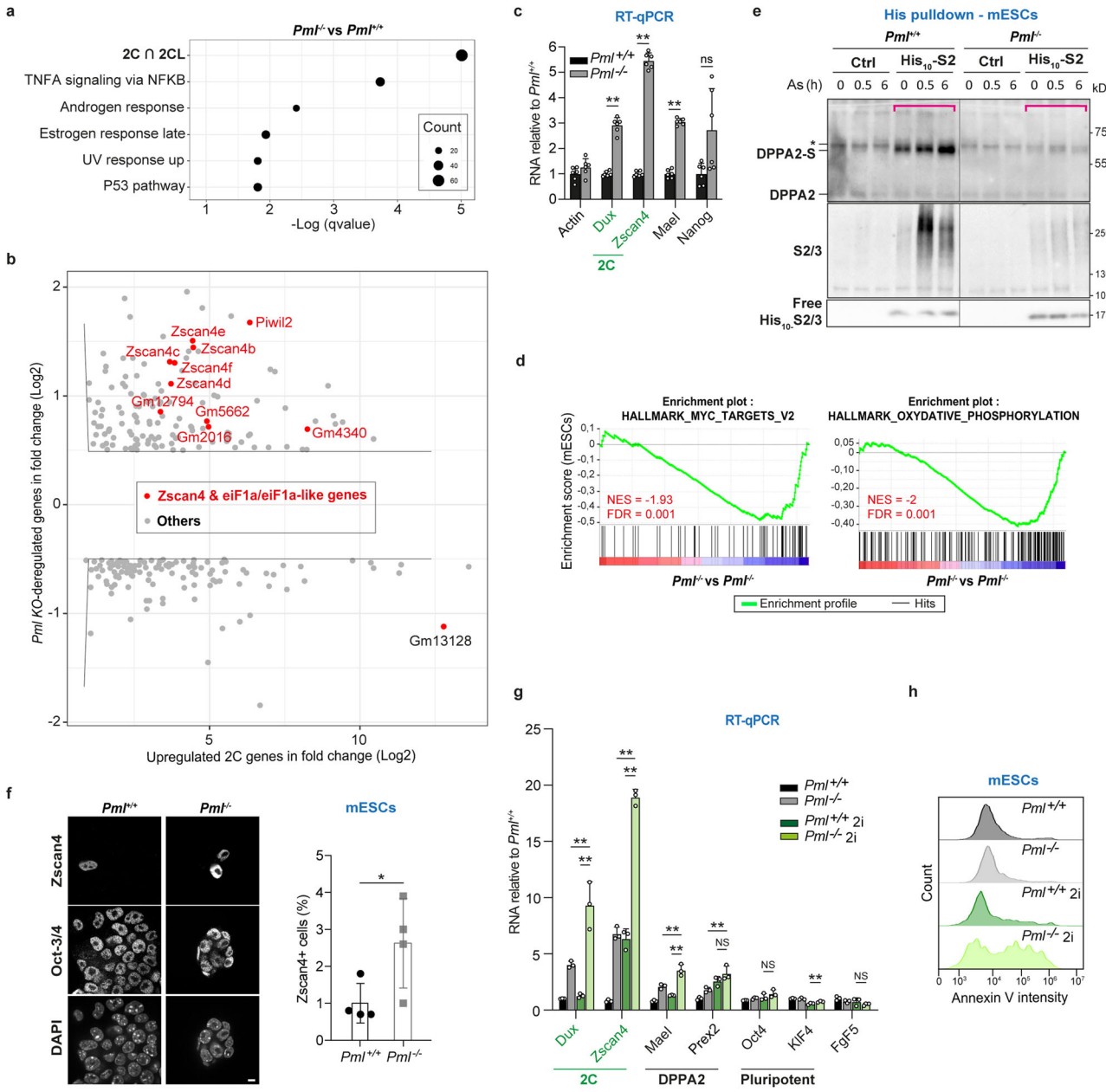

**Fig. 5 | PML is required for 2 CL transition and DPPA2 sumoylation in mESCs.**
**a** GSEA gene-set term enrichment analysis of the transcripts deregulated in *Pml⁻/⁻* vs *Pml⁺/⁺* mESCs from microarray analysis; 2C ∩ 2CL geneset: transcripts common to 2C embryo and 2CLcells from (ref. 49). **b** Transcripts de-regulated in *Pml⁻/⁻ vs Pml⁺/⁺* mESCs, related to upregulated genes in 2C embryo vs oocytes⁴⁹. Transcripts with significant changes (gray) and key 2CL genes (red) are shown from microarray analysis (Supplementary Data 6). **c** Mean fold increase of the indicated 2CL transcripts and MaeI as a direct target of DPPA2 (qRT-PCR) *n* = 6 independent biological replicates, normalized on gapdh and relative to values in *Pml⁺/⁺* mESCs, +/−SD, **p < 0.01, two-tailed unpaired Mann−Whitney test. Representative of the three CrispR/Cas9 *Pml⁻/⁻* clones. **d** GSEA analyses of RNAseq from *Pml⁻/⁻* vs *Pml⁺/⁺* mESCs unraveling Myc targets and Oxidative phosphorylation as the top two altered

pathways (Supplementary Data 6). **e** PML-dependent arsenic-induced S2-conjugation of DPPA2, PD from the indicated mESCs. Inputs in Supplementary Fig. 4h. Representative data, *n* = 5 independent biological replicates. **f** Representative image (left) and percentages of Zscan4+ cells (right), increased among *Pml⁻/⁻* mESCs, mean ± SD, *n* = 4 independent experiments; total of 400 (left) and 3000 (right) nuclei from *n* = 4, *p < 0.05, unpaired two-tailed Student's *t*-test. **g** Mean fold change of the indicated 2CL transcripts, DPPA2 target and pluripotent genes from the indicated mESCs exposed or not to 2i medium. *n* = 3 independent biological replicates, normalized on gapdh and relative to unpaired values in *Pml⁺/⁺* mESCs, +/−SD, **p < 0.01, two-tailed unpaired Mann−Whitney *t*-test. **h** FACS analysis to quantify Annexin V staining in *Pml⁻/⁻* vs *Pml⁺/⁺* mESCs exposed or not to 2i medium. Representative *n* = 3. Source data are provided as a Source Data file.

signals were sorted by flow cytometry. HeLa and mES cells were transfected with pBOS-KAP1-YFP plasmid using lipofectamine 2000 (Invitrogen, #11668019) and Lipofectamine 3000 (Invitrogen, #L3000001), respectively. KAP1-YFP positive mESCs were selected using 6 µg/ml basticidin (Thermo Fischer #R21001); and YFP + living cells were analyzed and sorted by flow cytometry (BD FACSAria2, v9.0.1), after FCS/SSC gating.

Cells were treated with 1 µM As₂O₃ (Sigma, #01969), 1 µM retinoic acid (Sigma, # R2625), or 1 µM ML792 (Medchemtronica # HY-108702) when requested.

**CRISPR/Cas9 knockout mESCs**
To knockout *Pml*, a single guide RNA targeting *Pml* exon 1 (Supplementary Data 7) was subcloned into the pSpCas9 (BB)−2A-Puro

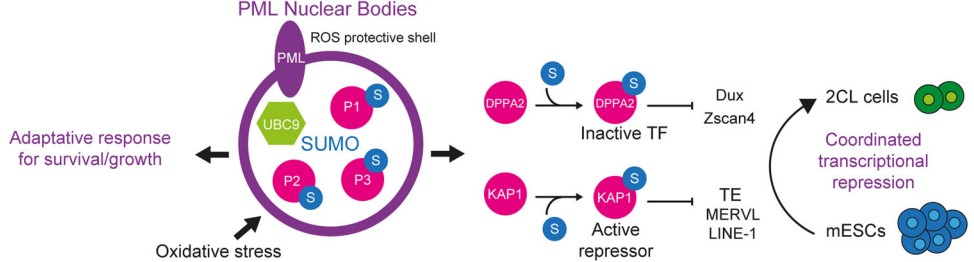

**Fig. 6 | Functional impact of PML NB-facilitated sumoylation.** PML NBs enhance the sumoylation of partner proteins (in pink). Through control of their sumoylation, PML coordinates the activity of KAP1 and DPPA2, orchestrating the inhibition of 2CL features. These key regulators of mESC fate silence ERVs and 2CL genes. Depending on the cell environment, PML NBs may also drive an adaptative stress response.

plasmid (pX459v2, Addgene), and transfection of mESCs was performed using lipofectamine 3000 (Invitrogen, #L3000001). Puromycin-resistant colonies were expanded and three clones deficient for *Pml* were chosen after DNA sequencing and Western blot analysis.

### SiRNA transfection

Mouse ESCs were transfected with siRNA (Dharmacon, Supplementary Data 7) with Lipofectamine 3000 (Invitrogen, #L3000001) according to the manufacturer's standard recommendation, and analyzed after 3 days.

### Mouse models and treatments

Animals were handled according to the guidelines of institutional animal care committees using protocols approved by the "Comité d'Ethique Experimentation Animal Paris-Nord" (no. 121). Mice were maintained in a 12 h light-dark cycle animal facility under specific pathogen-free conditions with free access to water and food (A03: SAFE; Institut de Recherche Saint Louis, Paris, France).

$His_6$-HA-*Sumo1* knock-in mice deficient for *Pml* were obtained after backcrossing of 129/sv *Pml*[-/-] mice (Pier Paolo Pandolfi, USA) with C57Bl/6 $His_6$-HA-*Sumo1* knock-in mice (Nils Brose, Netherland) for seven generations. Genotyping was performed by multiplex PCR using specific primers to distinguish *Pml*[-/-] (660 bp) from *Pml*[+/+] (443 bp) and tagged *Sumo1* (210 bp) from endogenous *Sumo1* (163 bp).

*Pml*[E167R/E167R] mice were obtained by CRISPR/Cas9 genome edition, performed on BALB/cByJ zygotes, using TAKE methods[81]. Briefly, three to four-week-old BALB/cByJ females were super ovulated using CARD HyperOva (Cosmo bio, #KYD-010-EX) and human Chorionic Gonadotropin, Sigma; #CG-10) and then mated with males (8–20 weeks) to get zygotes. crRNA, TracrRNA, ssDNA, and Cas9 nuclease were purchased from IDT and electroporated (NEPA21; Sonidal) to introduce *Pml* point mutation encoding E167R substitution (Supplementary Data 7). Genotyping performed as described above and PCR products were sequenced (Supplementary Data 7). BALB/c *Pml*[E167R/E167R] mice were then crossed with C57BL/6 *Sumo1*[His6-HA/His6-HA] mice to obtain *Pml*[E167R/-];*Sumo1*[His6-HA/wt] heterozygote mice.

For APL mouse model, leukemic blasts of derived from h-MRP8-*PML/RARA* transgenic mice[82] were transduced with MSCV-IRES-EGFP-$His_{10}$-SUMO2 or -$His_{10}$-SUMO2K/R and transplanted by intravenous (i.v.) injection into FVB mice. Serial i.v. transplantations were performed using $10^4$ GFP + sorted APL cells from bone marrow.

7–8 week-old age and sex-matched mice were used for all experiments; treatments with arsenic (5 μg/g, Sigma, #202673), pI:C (20 μg/g, invivogen, #tlrl-picw), or Bortezomib (1 mg/g, CliniSciences, #A10160-25) administered by intraperitoneal (i.p.) injection. Control mice were from Charles River or APL mice (untransduced blasts), same age, sex and substrains. The number of animals used are indicated in the figure legends.

### Constructs

Coding sequences for $His_{10}$-SUMO1, $His_{10}$-SUMO2, or all-lysine to arginine $His_{10}$-SUMO2 mutant ($His_{10}$-SUMO2K/R) were PCR amplified from MSCV-HA-SUMO1 or pLV-$His_{10}$SUMO2-Q87R or pVL-$His_{10}$-SUMO2-allKR-Q87R constructs[9,27] and cloned into MSCV-IRES-EGFP retroviral vector. $His_{10}$-encoding adaptor was cloned into MSCV-SUMO1-IRES-GFP for $His_{10}$-SUMO1construct (Supplementary Data 7). For MSCV-GRX1-roGFP2-SP100 construct, pGRX1-roGFP2 plasmid was used to insert GRX1-roGFP (Meng-Er Huang, Curie Orsay, France) into pMSCV-SP100 construct[9]. For pBOS-KAP1-YFP plasmid, KAP1 was PCR amplified from pEGFP-KAP1 (Addgene, #45568) and inserted in place of PMLIII into pBOS-PMLIII-YFP plasmid.

### Incucyte cell proliferation assay

To monitor cell growth upon arsenic, 5000 mESCs were seeded in triplicates on 96-well plates and analyzed with the IncuCyte live-cell imaging system (Essen). Four images per well were acquired with a 10× objective every 2 h for a period of 2–5 days. Sequential images were then analyzed using IncuCyte software.

### Immunofluorescence and Glutathione redox potential assay

Cells were fixed in PFA 3.7% (Sigma, #HT5011) for 15 min and permeabilized in PBS 0.2% Triton X-100 for 15 min. For frozen mouse liver, 10 μm sections were fixed for 15 min in PFA 3.7% and permeabilized in PBS 0.2% Triton-X100 for 15 min at RT. Incubation with primary or secondary antibodies was performed in PBS 0.2% Triton X-100 and 1% BSA for 2 h before staining with DAPI for 3 min. Primary antibodies: anti-Daxx (1/1000; cat#7152, Santa-cruz), anti-Oct4 (1/1000; Cat#Sc5279, Santa-Cruz), anti-PML(1/2000; Cat#MAB3738, Millipore), anti-SUMO-1 (1/1000; Cat#Ab3875, Millipore), anti-SUMO-2 (1/1000; Cat#Ab81371, Abcam), anti-UBC9 (1/1000; Cat# 610749, BD sciences), anti-Ubiquitin (1/1000; Cat#BML-PW8810, Enzo), anti-Zscan4 (1/1000; Cat# AB4340, Millipore). Secondary antibodies: anti-Rabbit Alexa Fluor 488 (1/200; Cat#A-11034, Sigma), anti-Mouse Alexa Fluor 594 (1/200;Cat#A-115-585-003, Jackson ImmunoResearch), anti-Rabbit Alexa Fluor 647 (1/200; Cat#A-111-605-003; Jackson ImmunoResearch). Proximity ligation assay was performed as previously described[9] with anti-E4BP4/NFIL3 (1/1000; Cat# 14312, Cell signaling), anti-PML(1/2000; Cat#MAB3738, Millipore) and anti-MORC3 (1/1000; Cat#NBP1-83036, Biotechne) antibodies. Image acquisitions were done by confocal microscopy (Spinning disk (CSU-WI, Yokogawa) and LSM 880 (Zeiss)) or by using an Axiovert-200 inverted-fluorescence microscope (Zeiss). Images were analyzed with FIJI software.

MEFs stably expressing GRX1-roGFP2 or GRX1-roGFP2-SP100 redox sensors were seeded in glass-bottom μ-dish (Biovalley). Measurement of the glutathione redox potential was performed by confocal microscopy analysis (Spinning disk (X1/TIRF) using 405-nm and 488-nm lasers. Ratios between quantified oxidized-405nm and reduced-488 nm forms of GRX1-roGFP2-SP100 were calculated within

regions of interest or the whole cell. Normalization was performed using 100 µM diamide (Sigma, #D3648)) and 5 mM DTT (Sigma, #D0632) treatments to determine the maximal and minimal oxidation capacities of the GRX1-roGFP2 sensor.

## Western blot and His pulldown

Whole-cell extracts were obtained from bone marrow, APL spleen, or cell lines, after washed in PBS 1×, by lysis and sonication in 2× Laemmli buffer (Sigma, #S3401). Frozen mouse livers were first homogenized using TissueLyser II (Qiagen), lysed in 1× Laemmli buffer and processed as above. 10–20 µg of proteins were loaded for SDS-PAGE on 4–12% or 4–20% NuPAGE Bis-Tris gels (Life Technology) and transferred to nitrocellulose membranes. Primary antibodies were incubated overnight at 4 °C and secondary antibodies 1 h at room temperature (RT). Antibodies used: anti-Actin (1/5000; Cat#A2066, Sigma), anti-DDPA2 (1/1000; Cat#MAB4356, Millipore), anti-HA (1/1000; Cat#901501; Bio-Legend), anti-MBP (1/1000; Cat#M3221; Sigma), anti-PML(1/1000; Cat#MAB3738, Millipore); anti-SUMO-1 (1/1000; Cat#AB3875, Millipore); anti-SUMO-2 (1/1000; Cat#Ab81371, Millipore), anti-TRIM28/Kap1 (1/1000; Cat#4124; Cell signaling), anti-UBC9 (1/1000; Cat#610749, BD Biosciences), anti-Ubiquitin (1/1000; Cat#BML-PW8810, Enzo); Secondary antibodies from Jackson ImmunoResearch: anti-Mouse-HRP (1/5000; Cat#115-035-062), anti-Rabbit-HRP (1/5000; Cat#111-035-045). Proteins were detected using Dura ECL or Super Signal West Femto (ThermoFisher), using Vilber Fusion-Fx (BIO-1D v15.07). Quantifications of proteins of interest (as indicated in the figures) were relative to loading control. All uncropped blots are provided within the Source Data file.

For His$_6$-HA-SUMO1 purification, frozen livers were homogenized as above and lysed with 6 M guanidine-HCl, 100 mM Na$_2$HPO$_4$/NaH$_2$PO$_4$ pH 8.0, 10 mM imidazole pH 8.0 and boiled at 95 °C for 5 min. After sonication, the protein concentration was determined using ™BCA Protein Assay kit (Thermo Scientific, #23225). Lysates were equalized and His$_6$-HA-SUMO1-conjugates were enriched on nickel-nitrilotriacetic acid (NiNTA) agarose beads (Qiagen, #L30210) as described in[15]. For dual purifications, the samples were then diluted with RIPA buffer, 0.4% NaDoc, 1% NP 40, 1% Triton X-100, 5 mM EDTA, pH 7.5, 20 mM NEM (Sigma, #E3876), PIC (Proteases Inhibitor Cocktail, Roche, #11836170001). Anti-HA immunoprecipication was additionally performed when indicated. Samples were incubated at 4 °C ON with anti-HA Affinity Matrix (Sigma, #11815016001), SUMO targets were eluted with HA peptide (Sigma, #11666975001). Pulldown of His$_{10}$-SUMO conjugates from APL cells or mESCs at small scale was performed as above with wash buffers containing 50 mM Imidazole. His$_6$-MBP recombinant protein was added to the samples as an internal control of the pulldown step efficacy, when possible.

For large-scale purification of His$_{10}$-SUMO2 conjugates from APLs, leukemic mouse spleens (with >70% of GFP+ leukemic cells) were dissociated in culture medium and washed in PBS. For SUMO proteomics in mESCs, around 200 million (GFP+) mESCs and four replicates for each condition were used. His$_{10}$-SUMO2 conjugates were purified as described in[27,83]. Briefly, cells were either lysed with Laemmli buffer as inputs or stored as dry frozen pellets. Guanidine lysis buffer (above) was added to frozen pellets for sonication and protein concentration was assessed using BCA Protein Assay Reagent (Thermo Scientific, #23225). Lysates were equalized for protein concentration and incubated with NiNTA agarose beads (Qiagen, #L30210) for O/N. Beads were washed using buffer 1 with 5 mM β-mercaptoethanol and 0.1% de Triton X-100), buffer 2 (8 M urea, 100 mM Na$_2$HPO$_4$/NaH$_2$PO$_4$ pH 8.0, 10 mM Tris pH 8.0, 10 mM imidazole pH 8.0, 5 mM de β-mercaptoethanol and 0.1% Triton X-100), buffer 3 (8 M urea, 100 mM Na$_2$HPO$_4$/NaH$_2$PO$_4$ pH 6.3, 10 mM Tris pH 6.3, 10 mM imidazole pH 7.0 and 5 mM de β-mercaptoethanol) and with buffer 3 without imidazole. Samples were eluted in 7 M urea, 100 mM Na$_2$HPO$_4$/NaH$_2$PO$_4$, 10 mM Tris pH 7.0 and 500 mM imidazole

pH 7.0. Eluates were passed through pre-washed 0.45µm filter columns (Millipore) to remove any residual beads and subsequently concentrated on pre-washed 100 kDa cut off filters (Sartorius). Sample volume were equalized to 50 µL (5% of the samples were used as pulldown control for Westen blots analysis) and digested by LysC (Wako, #129-02541) and Trypsin (Promega, # V5111), and acidified using 2% TCA (Sigma, # T1647). Peptide samples were loaded on C18 StageTips and dried using vacuum.

## Mass spectrometry data acquisition

Both His$_{10}$-SUMO conjugates were processed by mass spectrometry analyses in Alfred Vertegaal laboratory (LUMC, Netherland). All the experiments were performed on an EASY-nLC 1000 system (Proxeon, Odense, Denmark) connected to a Q-Exactive Orbitrap (Thermo Fisher Scientific, Germany) through a nano-electrospray ion source as previously described[18]. The Q-Exactive was coupled to a 15 cm analytical column with an inner diameter of 75 µm, in-house packed with 1.9 µm C18-AQ beads (Reprosher-DE, Pur, Dr. Manish, Ammerbuch-Entringen, Germany). The gradient length was 120 min from 2 to 95% acetonitrile in 0.1% formic acid at a flow rate of 200 nL/minute. For the samples from cohort 2 and 3, two technical repeats were performed and the mass spectrometer was operated in data-dependent acquisition mode with a top 5 method. Full-scan MS spectra were acquired at a target value of $3 \times 10^6$ and a resolution of 70,000, and the Higher-Collisional Dissociation (HCD) tandem mass spectra (MS/MS) were recorded at a target value of $1 \times 10^5$ and with a resolution of 17,500 with a normalized collision energy (NCE) of 25%. The maximum MS1 and MS2 injection times were 20 and 250 ms, respectively. The precursor ion masses of scanned ions were dynamically excluded (DE) from MS/MS analysis for 20 s. Ions with charge 1, and >6 were excluded from triggering MS2 analysis. For cohort 1, the mass spectrometer was operated in data-dependent acquisition mode with a top 7 method, two independent technical repeats were performed. Full-scan MS spectra were acquired at a target value of $3 \times 10^6$ and a resolution of 70,000, and the Higher-Collisional Dissociation (HCD) tandem mass spectra (MS/MS) were recorded at a target value of $1 \times 10^5$ and with a resolution of 35,000 with a normalized collision energy (NCE) of 25%. The maximum MS1 and MS2 injection times were 20 and 120 ms, respectively. The precursor ion masses of scanned ions were dynamically excluded (DE) from MS/MS analysis for 60 s. Ions with charge 1, and >6 were excluded from triggering MS2 analysis.

## Mass spectrometry data analysis

All RAW data were analyzed using MaxQuant (version 1.5.3.30) according to ref. 84. We performed the search against an in silico-digested UniProt reference proteome for Mus musculus (24 March 2016) and the human PML/RARA protein.

Database searches were performed with Trypsin/P, allowing four missed cleavages. Oxidation (M) and Acetyl (Protein N-term) were allowed as variable modifications with a maximum number of 5. Match between runs was performed with 0.7 min match time window and 20 min alignment time window. The maximum peptide mass was set to 5000. Label-Free Quantification was performed using the MaxLFQ approach, not allowing Fast LFQ[85]. Instrument type was set to Orbitrap.

Protein lists generated by MaxQuant were further analyzed by Perseus (version 1.5.5.3)[86]. Proteins identified as common contaminants were filtered out, and then all the LFQ intensities were log2 transformed. Different biological repeats of the experiment were grouped and only protein groups identified in all biological replicates in at least one group were included for further analysis. Missing values were imputed using Perseus software by normally distributed values with a 1.8 downshift (log2) and a randomized 0.3 width (log2) considering whole matrix values.

Proteins were considered to be SUMO2 targets when the difference between His$_{10}$-SUMO2 APL samples and their respective (Ctrl)

APL control samples were statistically significant for $p < 0.05$ (t-test) and larger than +0.7 (log2). Thus, total SUMO targets identified in each cohort were obtained by adding specific targets found in $His_{10}$-S2 APL *vs* Ctrl APL mice and arsenic-treated $His_{10}$-S2 APL *vs* arsenic-treated Ctrl APL mice. Then, we excluded proteins which total amount might increase upon arsenic, based on their increase upon arsenic in the Ctrl APL background, by comparing LFQ FC in Ctrl APL mice treated or not with arsenic.

We used a similar strategy for $His_{10}$-S2 mESCs and Ctrl mESCs, with $Pml^{+/+}$ or $Pml^{-/-}$ genotypes, treated with 2 h of $As_2O_3$ 1 μM. We selected proteins as being SUMO2 targets when the difference between $His_{10}$-SUMO2 mESCs samples and their respective Ctrl mESC samples (with the same genotype) was statistically significant for $p < 0.05$ (FDR) and larger than +0.7 (log2).

Comparison with PML interactant was made using the Biogrid database [https://thebiogrid.org/111384/summary/homo-sapiens/pml.html]

### Chromatin immunoprecipitation
ChIP was performed with the iDeal ChIP-qPCR kit (Diagenode, #C01010180) according to the recommendation of the manufacturer. Briefly, mESC cells were crosslinked in 1% formaldehyde (Euromedex, #EM-15686) for 15 min at room temperature and quenched with 0,12 M glycine provided in the kit. The extracted chromatin was sonicated with a Bioruptor Pico (Diagenode) and immunoprecipitation was performed using 1ug of an anti-H3K9me3 (Abcam, #ab8898) and 1ug of a rabbit IgG-isotype (Diagenod, #C15410206) control antibody. Eluted DNA was quantified by real-time PCR on a Roche LightCycler by using FastStart Universal SYBR Green Master (Roche, #4913850001) with the set of primers listed in Supplementary Data 7. Data were normalized with respect to percentage of input and correspond to the mean +/− SD from at least three replicates.

### RNA preparation and quantitative RT-PCR
Total RNA extraction from mESCs was performed with the RNeasy Plus Mini Kit (Qiagen, #74134) with an additional in-column DNase treatment. RNA was quantified using a Nanodrop One-One (Thermofisher) before cDNA amplification or RNA seq libaries preparation. cDNA were prepared from 1 μg of total RNA with a Maxima First Strand cDNA Synthesis kit (Thermofisher, #1641) including an additional step of DNAse treatment before reverse transcription. Quantitative real-time PCR on cDNA was then performed as described for ChIP experiment. Expression levels were normalized to endogenous *Actin* or *Gapdh* gene as indicated.

### RNA sequencing
Quality control of purified RNA was performed using a 2100 BioAnalyzer (Agilent). Sequencing libraries were prepared with 800 ng of total RNA by using TruSeq Stranded Total RNA Ribo-Zero Gold Prep kit (Illumina, #20020598) allowing depletion of cytoplasmic and mitochondrial rRNA. Briefly, after reduction of rRNA with target-specific oligos combined with Ribo-Zero rRNA removal beads, purifed RNA was then fragmented and sequentially reverse-transcribed with random primers into double-stranded cDNA fragments. After adapter ligation, cDNA fragments were enriched by PCR to obtain barcoded libraries size-selected with AMPureXP beads (Beckman Coulter, #A63881). Quantification and quality control of each library was assessed by using a 2100 Bioanalyzer. Final libraries were then normalized and pooled in equal molar concentration for 75 bp single-end sequencing with an Illumina NextSeq 550.

### RNA-SEQ mapping
The sequencing reads raw data (FASTQ) were submitted to quality check and were trimmed for Illumina adapters sequences and low-quality bases using Trim Galore (version 0.4.5 with default parameters, [http://www.bioinformatics.babraham.ac.uk/projects/trim_galore/]). Trimmed read pairs (>35 bp) were mapped to mouse reference genome (mm10 [https://www.ncbi.nlm.nih.gov/assembly/GCF_000001635.20/]) using STAR[87] aligner (version 2.5.2b) with maximum multiple alignments of no more than 100, using the variables -*winAnchorMultimapNmax 100* and −*outFilterMultimapNmax 100*. From BAM files, TEtranscripts[88] (version 2.0.3) was used to quantify both gene (uniquely aligned reads only) and transposable element transcript abundances (including both unique- and multi-aligned reads). The differential expression analysis was done by using the DESeq2 package[89] for modeling the counts data with a negative binomial distribution and computing adjusted *P*-values. For the comparative analyses ($Pml^{-/-}$ versus WT), only genes with both FC > 2 and FDR < 0.05 were considered as differentially expressed.

### Microarrays transcriptomic data analysis
Total RNA was extracted from mESCs as described for RNA sequencing analysis. Affymetrix GeneChip® Mouse Transcriptome Assay 1.0 (MTA 1.0) was used to perform gene expression analysis on 3 $Pml^{-/-}$ and 3 $Pml^{+/+}$ mESC samples. Background correction, probe set signal integration, and quantile normalization were performed through Robust Multichip Analysis (RMA) algorithm, as implemented in the "R" Affymetrix package. Genes whose abs(log2(fold-change)) between comparative group was ≥0.5 (with a *p* value cut-off of <0.1) were selected. In addition, false discovery rate (FDR) was applied for multiple hypotheses testing using Benjamini-Hochberg correction. Genes with a FDR-adjusted *p* value (adjusted *p* value) ≤0.05 were finally accepted. Biological gene-pathway changes were scored using the Gene Set Enrichment Analysis algorithm (GSEA) using hallmark signature database (MSigDB, [http://www.gsea-msigdb.org/gsea/msigdb]). A custom specific 2 C∩2CL pathway was used based on[49] for transcripts restricted to both 2-cell embryo and 2CL cells.

### Statistical analysis
The number of independent experimental replications is indicated in the legends. Statistical analyses of mean and standard deviation were performed with Prism 7 (GraphPad Software) as well as Student's *t*-test, Welch or Wilcoxon Mann-Whitney tests; FDR for multiple hypotheses testing using Benjamini-Hochberg correction, as indicated.

### Reporting summary
Further information on research design is available in the Nature Research Reporting Summary linked to this article.

## Data availability
The SUMO proteomic data generated in this study have been deposited in the ProteomeXchange Consortium, PRIDE[90] partner repository database, for APL data under the accession number PXD019609; for the mESCs data under the accession number PXD028865. The RNA seq data generated in this study have been deposited the EMBL-EBI database [https://www.ebi.ac.uk/] under the accession number E-MTAB-10153 and Affimetrix Microarrays transcriptomic data under the accession number E-MTAB-10151. Source data are provided with this paper.

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

## Acknowledgements

This work was supported by grants from Agence Nationale pour la Recherche, ANR SUMOPiv (V.L.B.); the ERC, PML-Therapy

(ADG-785917) (H.D.T.); Fondation ARC (V.L.B.); ANRJC (P.T.). We kindly thank the staff of the animal facility in Research Institute of Saint–Louis (IRSL), A.L. Maubert for her help with mouse care. We are also grateful to image core microscopy facilities of IRSL and College de France, Paris, in particular N. Setterblad for his helpful advice with incucyte analysis. We warmly thank M. Tirard and N. Brose (Max Planck Institute of Experimental Medicine, Göttingen, Germany) for the sharing of His6-HA-*Sumo1* knock-in mice, and Meng-Er Huang (Institut Curie, Orsay, France) for his GRX1-roGFP2 construct. We thank Pablo Navarro (Pasteur) for providing mESCs. We also acknowledge P. Mayeux and V. Salnot of the Proteomic 3P5 platform in Cochin hospital, Paris, France, for the total APL proteomics. The Orbitrap Fusion mass spectrometer was acquired with funds from the FEDER through the « Operational Programme for Competitiveness Factors and employment 2007-2013 », and from the « Cancéropôle Ile-de-France ». We kindly thank P. Lesage, A. Amara, M.H. Verhlac and D. Bourch'is for their critical readings of the manuscript, and other members of the team for helpful advice. We finally thank the support services of IRSL and CIRB.

## Author contributions

S.T., O.F., M.C.G., performed all experiments; R.G.P. performed SUMO proteomics (quantitative MS/MS analyses). D.R. extracted the transcripts for Affymetrix microarray. S.Q. analyzed RNAseq and transcriptomics data, A.C., E.F. and P.T. generated the *Pml*$^{-/-}$ mESCs, MNK the *Pml*$^{E167R/E167R}$ and *His$_{10}$-HA-Sumo3* knock-in mouse, M.P. for *Pml*$^{-/-}$ and *Pml*$^{E167R/E167R}$ mouse crossing with His$_6$-HA-*Sumo1* knock-in mice. M.T. generated the His$_6$-HA-*Sumo1* Knock-in model[28]. P.B. generated Grx1-roGFPconstructs and cell experiments. O.F., M.C.G., A.V., V.L.B. and H.d.T. designed experiments, interpreted data and contributed to the writing of the manuscript. All authors reviewed the manuscript.

## Competing interests

The authors declare no competing interests.

## Additional information

[1]Center for Interdisciplinary Research in Biology (CIRB), Collège de France, PSL research university, Inserm, Cnrs, 11 place Marcelin Berthelot, 75005 Paris, France. [2]Université Paris Cité, Inserm, CNRS, GenCellDis, Institut de Recherche Saint-Louis, F-75010 Paris, France. [3]Department of Cell and Chemical Biology, Leiden University Medical Center (LUMC), Einthovenweg 20, 2333, ZC Leiden, The Netherlands. [4]Service de Hématologie, AP-HP, Hôpital St. Louis, F-75010 Paris, France. [5]Normal and pathological hematopoiesis: Emergence, Environment and translational research, Université de Paris Cité, Inserm, IRSL, F-75010 Paris, France. [6]Department of Molecular Neurobiology, Max Planck Institute of Multidisciplinary Sciences, D-37075 Goettingen, Germany. [7]Present address: Honing Biosciences, Hôpital Saint Louis, 16, rue de la Grange aux Belles, 75010 Paris, France. [8]Present address: Genome Proteomics laboratory, Department of Cell Biology, Andalusian Center for Molecular Biology and Regenerative Medicine (CABIMER), University of Seville, Seville, Spain. [9]Present address: Institute of Epigenetics and Stem Cells (IES), Helmholtz Zentrum München, Feodor-Lynen-Straße 21, 81377 München, Germany. [10]These authors contributed equally: Sarah Tessier, Omar Ferhi, Marie-Claude Geoffroy. ✉e-mail: hugues.dethe@inserm.fr; valerie.lallemand@inserm.fr

