## [Peer Review File · Nature Communications]

Exploration of nuclear body-enhanced sumoylation reveals that PML represses 2-cell features of embryonic stem cellsREVIEWER COMMENTS

Reviewer #1 (Remarks to the Author):

In their manuscript Tessier et al. study the effects of PML-driven modification by Sumo2/3 employing expression of His-tagged sumo proteins in 3 experimental systems: 1. mESC (Crispr-PMI KO), 2. APL mice and 3. Livers from PML-/-X His-sumo mice. Arsenic acid was used as a booster of sumoylation. The authors have identified PML-dependent sumo 2/3 targets employing LFQ LC-MS/MS on 3 cohorts of APL mice treated by arsenic acid (1, 3, 6 hrs). Among their targets they focus on the KAP1 repressor complex and continue their further studies on mouse ESC. They show de-repression of endogenous retroviruses (ERVs)- KAP targets- in the absence of PML and perform a transcriptomic (Affymytrix+ RNA-seq) in ESC WT and PML-/. Their final conclusion is that, by sumoylating KAP1 and Dppa2, PML opposes the conversion of ESC into 2C-cell state. The submitted work is interesting to a wide research community in the fields of stem cell potency, oxidative stress response and sumoylation-dependent repression of transcriptional activators and repressors. The manuscript describes in an explicit manner a big amount of data! The experiments are well executed, controlled and interpreted. The conclusions are strong and reliable.

However the work suffers from one weakness that concerns the originality:

1. A previous publication has reported the importance of sumoylation in the maintenance of naïve ES cells by preventing reversion into the 2C-like state (Cossec et al., 2018, Cell Stem Cell 23).
2. In another publication PML and PML-nuclear bodies were shown to regulate the transition between ESC and 2C-line cells (Kurihara et al., 2020, Molecular Cell 78).
3. Lastly it was recently reported that SUMOylation of Dppa2/4 prevents conversion of ESCs to the 2C-like state (Theurillat et al., 2020, Cell Reports 32).

In order to increase the impact of the publication the authors should examine thoroughly the mechanism whereby PML-derived sumoylation of distinct factors inhibits the ESC to 2C-like conversion.

For example, concerning the KAP1 factor, the authors should use a WT along with a sumoylation-deficient mutant in a ESC line that is KAP1 -/- and PML-/- in order to clearly demonstrate that it is PML-driven sumoylation of KAP1 that restrains the ESC to 2C conversion.

Reviewer #2 (Remarks to the Author):

PML nuclear bodies represent a distinct subnuclear domain composed of multiple proteins. The cellular function of PML nuclear bodies is still not entirely clear, but circumstantial evidence indicates that these structures are centers for modification with the ubiquitin-related SUMO molecule and stress-induced ubiquitylation of polySUMOylated proteins. Previous work by de The and Lallemand-Breitenbach (Sahin et al., JCB, 2014) proposed that oxidative stress-induced assembly of PML NBs controls SUMOylation of partner proteins. In the current manuscript the authors aimed to identify the PML NB-dependent SUMO targets by combining cell-based and in vivo experiments. Following the identification of potential PML NB-dependent SUMO conjugates the authors concentrate on KAP1/TRIM25 in

subsequent functional studies and propose that PML controls the transition of ESCs to the 2-cell like state by enhancing SUMOylation of KAP1 and other key regulators in this pathway. These findings connect published work demonstrating that both SUMO and PML function as important regulators of stem cell pluripotency (Hadjimichael et al., Stem cell reports, 2017; Cossec et al., Cell Stem Cell, 2019).

To my opinion, the manuscript contains a wealth of data that clearly provide several interesting novel aspects, including the identification of candidate PML-dependent SUMO targets and the unravelling of PML's role in orchestrating ESC fate. However, as outlined in detail below some important aspects of the manuscript are not yet fully conclusive. In particular, I feel that the APL model system is not ideal for the identification of PML NB-dependent SUMO substrates by proteomics. Further, the direct link of PML-enhanced SUMOylation to mESC transition is missing. Lastly, the proposed shielding of Ubc9 from oxidation in PML NBs is an appealing concept, but at this stage the data supporting the model are rather preliminary. Generally, the manuscript is not very easy to read and the authors occasionally take argumentative short-cuts to support their claims.

1) Based on the experiments in Figure 1 it is concluded that PML NBs mediate basal and stress-induced SUMO conjugation. However, to my opinion, these data are not fully conclusive for several reasons. First, it is concluded that PML controls both basal and stress-induced SUMOylation. The interpretation relies on the comparison of the general SUMOylation pattern observed by immunoblotting in PML^{-/-} mESCs and the corresponding control cells (Figure 1D). There is indeed a huge difference when comparing SUMO2 conjugates in PML^{-/-} and PML^{+/+} mESCs. However, can the authors exclude that the reduced SUMOylation pattern primarily results from the lack of PML itself, which likely represents one of the most abundant SUMO2 target in cells. In line with this possibility PML-SUMO2 conjugates detected in PML^{+/+} cells (Figure 1d, right panel) migrate at exactly the molecular weight of the major SUMO-conjugates. Accordingly, these conjugates are lost in PML^{-/-} cells. The impression that PML controls the bulk of basal SUMOylation may therefore be somewhat misleading, since most targets are simply not detected by immunoblotting. The experiments on KAP1 in PML^{-/-} and PML^{+/+} mESCs (Figure 4B) support this view. The data rather show that polySUMOylation (or formation of hybrid SUMO/Ub chains), in particular under stress requires PML.

Second, the data obtained in the APL system are difficult to interpret with respect to the role of PML NBs in basal SUMOylation. The authors argue that in the PML-RAR expressing APL mouse model SUMOylation is impaired due to the loss of PML NB integrity. They further conclude that the restoration of SUMOylation by arsenic treatment is due to the re-assembly of PML NBs. However, the data obtained in the His-SUMO2K/R expressing mice (Figure 1e) are difficult to reconcile with this interpretation. As shown here, robust SUMOylation by His-SUMO2K/R is detected even in the absence of arsenic indicating that SUMOylation occurs very efficiently even when aberrant PML NBs are present in APL cells. Therefore, monoSUMOylation does not seem to be affected in the APL mice. In line with this interpretation the MS data suggest that core NB proteins, such as Sp100 exhibit robust SUMOylation in APL cells even without addition of arsenic. The APL model is therefore not the ideal system to explore PML NB-dependent basal SUMOylation.

2) Data provided in Supplementary Tables 1 +2 do not fully support the conclusion that the proteomics experiment (+/-arsenic) in the APL mice specifically identifies PML NB-dependent SUMO targets. I do acknowledge that a subset of arsenic-induced targets is connected to PML NBs, but the most strongly regulated proteins (e.g. YEATS2, BAZ1B) are -at least to my knowledge-not physically connected to PML or PML NBs. I actually do think

that the dataset presented here is very valuable and important to better understand SUMO dynamics in APL cells in response to arsenic treatment. However, at least to my opinion, the experimental setup does not properly reflect the title of the manuscript ("unbiased in vivo exploration of nuclear bodies-enhanced sumoylation...").

3) Following the identification of potential PML NB-dependent SUMO conjugates the authors concentrate on KAP1/TRIM25 in subsequent functional studies. The data support the conclusion that PML favors KAP1 SUMOylation (at least stress-induced polySUMOylation). Previous work has established that KAP1 SUMOylation is involved in transcriptional repression and in particular silences expression of endogenous retrovirus in embryonic stem cells (Yang et al., Cell 2015). Interestingly, the authors show here that PML deficiency also upregulates KAP-1-repressed ERVs possibly connecting PML to this pathway. However, what is missing here is a clear demonstration that PML exerts this effect via enhancing SUMOylation of KAP-1. Chemical or genetic inactivation of SUMOylation is needed to strengthen this point.

4) In the last part of the MS the authors show that PML controls the transition state of ESCs to the 2-cell like state. The transcriptome analysis of PML^{-/-} vs. PML^{+/+} mESCs supports this view and provides a mechanistic basis for this phenotype. Further, it is shown that in PML^{-/-} cells SUMOylation of key regulators of the 2-cell like stage, including DPP2/4 is reduced. Here, the authors need to refer to a recent paper by Theurillat et al. (Cell reports 2020), where the pluripotency factors Dppa2 and Dppa4 were identified as the top-scoring SUMO substrates in ESCs with SUMOylation of Dppa2 and Dppa4 impeding the conversion to 2-cell-embryo-like states.

5) The model that Ubc9 is shielded from oxidation in PML NBs is very speculative at this stage, since the underlying data are rather weak. Where was the PMLE167^{-/-} knock-in allele characterized (Figure 2A)? I was unable to find any information in the references that were provided. If no characterization is available it needs to be shown that PMLE167^{-/-} indeed affects partner recruitment. Further, why was the GRX-roGFP reporter fused to Sp100 and not to directly to Ubc9? Moreover, the difference in the bar graph is rather insignificant (Figure 2B). Generally, this aspect of the manuscript seems quite unrelated to the rest.

Additional points:

1. Fig 1a. Loading control for the SUMO2/3 blot is missing. Is it re-probed?
2. Fig 1b. In the chart on Figure 1b colored lines are not labeled, so it is unclear which line correspond to the respective treatments.
3. Fig. 1d. What is the significance and rationale of probing His-SUMO1 pull-down membrane with SUMO2/3 and vice-versa? Do the SUMO2/3 blot in the left panel and SUMO1 blot in the right panel indicate abundance of SUMO1+SUMO2/3 conjugated proteins? If so, why is there a difference between the SUMO2/3 blot in the left panel and SUMO1 blot in the right panel?
4. Page 5, 3rd line from the top. It is not clear what the authors want to say by "Moreover, in untreated Pml^{-/-} mESCs, basal His10-SUMO2 conjugation....by enhanced SUMOylation"? A better explanation would help.

5. The phrase “Basal levels of global SUMO2 conjugates... catabolism” (Page 5, 6th line from the bottom) is not clear.

6. Fig. 1f. What is the rationale of SUMO2/3 blot here in the His-SUMO1+HA pull down assay? Why is there a difference (which looks significant) between the 6th and 7th lanes from the left in SUMO2/3 blot, whereas the corresponding HA blot does not show any change?

7. Fig. 1g. Again, the significance of SUMO1 blot is not clear. Does it represent proteins which are conjugated to both SUMO1 and SUMO2/3? If so, the statement “Preliminary experiments in HA-His10-SUMO3 knock-in mice....SUMO1” should be changed.

Reviewer #3 (Remarks to the Author):

In this manuscript, Tessier et al. investigate the role of PML nuclear bodies (NB) in the regulation of embryonic stem cell fate. Using mESCs and mice deficient for PML or PML NB formation (APL mice), combined with expression of His-tagged SUMO variants, the authors demonstrate PML-dependent SUMO conjugation in vitro and in vivo and show that this correlates with stress-induced growth arrest of mESCs. They further show that PML-dependent SUMO conjugation is associated with ubiquitination and degradation of SUMOylated target proteins. Proteomics identification of SUMOylated proteins in APL mice expressing His-tagged SUMO2 revealed known NB components and members of the KAP1 chromatin remodeling complex as SUMO substrates. KAP1 is a master epigenetic regulator known to be involved in stem cell maintenance. The authors validated PML-dependent SUMOylation of KAP1 in mESCs and show that this is associated with PML-dependent repression of transposable elements and the conversion of mESCs into the 2-cell like (2CL) state.

The manuscript is well written and properly constructed, and the experimental data is of high quality, but I do have a few major concerns that should be addressed:

1) The overall coherence between the different parts and experiments could be improved. Many experiments show interesting associations, but do not show causality, providing rather limited mechanistic insight and often making it difficult to draw strong conclusions. For instance, to show that PML controls the mESC 2CL state via KAP1 SUMOylation, could the authors mutate individual sites in KAP1 or generate a SUMO death mutant of KAP1?

2) Regarding the proteomics results in Figure 3:

- Please clearly state in the main text which mouse tissue/organ was analyzed and why this sample type was chosen with regard to mESC biology.

- If I understand correctly, within each cohort 3 comparisons were made, with the first 2 comparisons (un His10-S2 APL versus un Ctrl APL; As His10-S2 APL versus As Ctrl APL) used to select SUMOylated proteins. Selected hits were then used for the 3rd comparison (As His10-S2 APL versus un His10-S2 APL) and plotted in the volcano plots shown in Fig

3b, to see how these SUMOylated proteins have an increase or decrease in SUMOylation upon arsenic acid treatment. This strategy should be clearly described in the main text.

- I understand that the total proteome was also measured and that the same comparisons were made to calculate protein fold change ratios between the different conditions. Please include an additional Table with the fold change ratios of the shotgun experiments and also add these protein fold change ratios for each SUMO target in Table 1. These data is currently represented in Supplementary Figure 3, but the heatmap shown there does not allow to lookup fold change ratios for individual proteins. This is particularly important since SUMO2-triggered degradation of proteins is expected over time.

- Table 1: please include a significance indication for each protein and each comparison. That will help the reader to understand the selection strategy and is also important to have a complete view of significantly up- or downregulated SUMO targets shown in Fig 3b. At present, it is for instance unclear what are the 19 targets with increased SUMOylation one hour after injection.

- Table 2: for clarity, please also include a sheet that represents the 90 targets mentioned in the main text. Now these targets are distributed over the three sheets, which makes it difficult to follow the behavior of one target over time.

Minor Comments:

- Figure 2A: this panel would fit better in Figure 1 and also the corresponding paragraph in the main text would fit better with the previous section.

- Materials and methods, mass spectrometry data acquisition: please correct sentence at line 9 and 10 of this paragraph

- Please clarify why the number of missed cleavages is set at an unusually high value of four during the database search.

REVIEWER COMMENTS

Reviewer #1 (Remarks to the Author):

In their manuscript Tessier et al. study the effects of PML-driven modification by Sumo2/3 employing expression of His-tagged sumo proteins in 3 experimental systems: 1. mESC (Crispr-PMI KO), 2. APL mice and 3. Livers from PML^{-/-}X His-sumo mice. Arsenic acid was used as a booster of sumoylation. The authors have identified PML-dependent sumo 2/3 targets employing LFQ LC-MS/MS on 3 cohorts of APL mice treated by arsenic acid (1, 3, 6 hrs). Among their targets they focus on the KAP1 repressor complex and continue their further studies on mouse ESC. They show de-repression of endogenous retroviruses (ERVs)- KAP targets- in the absence of PML and perform a transcriptomic (Affymetrix+ RNA-seq) in ESC WT and PML^{-/-}. Their final conclusion is that, by sumoylating KAP1 and Dppa2, PML opposes the conversion of ESC into 2C-cell state. The submitted work is interesting to a wide research community in the fields of stem cell potency, oxidative stress response and sumoylation-dependent repression of transcriptional activators and repressors. The manuscript describes in an explicit manner a big amount of data! The experiments are well executed, controlled and interpreted. The conclusions are strong and reliable.

We warmly thank this reviewer to underscore the interest of our findings for a wide scientific readership, and the solidity of our conclusions based on large reliable scientific data.

However, the work suffers from one weakness that concerns the originality: 1. A previous publication has reported the importance of sumoylation in the maintenance of naïve ES cells by preventing reversion into the 2C-like state (Cossec et al., 2018, Cell Stem Cell 23).

One could be under the impression that the links between PML, SUMO and 2CL state are not novel, due to prior reports. However, as detailed below, while these reports link 2CL state to SUMO or explore PML role in mESC, none of them actually mechanistically link the 3 fields. Thus, these publications rather highlight the timeliness of our studies.

Sumoylation was indeed shown to maintain mESCs identity (all these publications were cited in the main text, including Cossec et al. 2018). Yet, our study identifies PML as an upstream control of 2CL state through sumoylation, which is a complete novelty. The fact that these membrane-less organelles biochemically control various key effectors of 2CL, and that PML controls ERV expression in mESCs have never been proposed.

More broadly, our study establishes that PML Nuclear Bodies regulate the sumoylation of multiple novel NB targets identified by *in vivo* mass spectrometry, in particular regulators of ERV repression and 2CL fate in mESCs.

2. In another publication PML and PML-nuclear bodies were shown to regulate the transition between ESC and 2C-line cells (Kurihara et al., 2020, Molecular Cell 78).

This publication indeed deals with the role of PML in mESCs. However, it does not address the role of PML in 2CL transition nor on ERV repression. Rather, it assesses the chromatin status when proximal to PML NBs. In fact, it identifies PML binding onto

actively transcribed genes, notably within a large region on the Y chromosome. It is therefore completely unrelated to our own results.

3. Lastly it was recently reported that SUMOylation of Dppa2/4 prevents conversion of ESCs to the 2C-like state (Theurillat et al., 2020, Cell Reports 32).

We thank the reviewer for pointing out the missing reference by Theurillat et al. in Cell report 2020. We apologize for this omission which was corrected (p12, ref: 53).

In the two cited studies, DPPA2/4 sumoylation does not involve the PML. Our aim is not to re-demonstrate that the sumoylation of DPPA2/4 dimer inhibits its activity, but rather that the sumoylation activity of PML NBs controls the function of downstream target proteins. This allowed us to unravel a PML-dependent coordinated repression of retroelements and 2CL genes in mESCs.

In order to increase the impact of the publication the authors should examine thoroughly the mechanism whereby PML-derived sumoylation of distinct factors inhibits the ESC to 2C-like conversion. For example, concerning the KAP1 factor, the authors should use a WT along with a sumoylation-deficient mutant in a ESC line that is KAP1 $-/-$ and PML $-/-$ in order to clearly demonstrate that it is PML-driven sumoylation of KAP1 that restrains the ESC to 2C conversion.

We are grateful to the reviewer for her/his suggestion helping us to propose an improved version of our manuscript. KAP1 drives ERV repression^{1,2}, with key role of KAP1 sumoylation^{3,4}. However, we agree that we had not formally demonstrated that the phenotype of *Pml* knockout mESCs was due to defect of PML-driven KAP1 sumoylation. Focusing on KAP1 sumoylation and ERV regulations, we now provide conclusive results mechanistically linking PML to KAP1 sumoylation (see text p11, Figures 4g-j).

KAP1 knockout being lethal in mESCs⁵, we could not develop a full inactivation strategy and then introduce an expression vector in which KAP1 is devoid of its 6 mapped sumoylation site. Similarly, a knock-in strategy to mutate the 6 sites in both *Kap1* alleles of mESCs would be unrealistic. We have therefore used *Kap1* siRNA and first confirmed that both *Kap1* and *Ubc9* knockdown (KD) led to ERV de-repression, similar to *Pml* knockout (although of smaller magnitude) (new Fig. 4g). We then explored the effects of *Kap1* KD in a *Pml* KO background and demonstrate that in cells where ERVs were already derepressed, *Kap1* siRNA did not further enhance expression of endogenous retroviruses (new Fig. 4h). Then, as suggested by Reviewer 2, we used the ML792 inhibitor to block sumoylation. Critically, while ML792 enhanced ERV expression in *Pml*^{+/+} cells, it had no significant effects in *Pml* KO mESCs (new Fig. 4i, supplementary Fig. 4d). Finally, we could restore the repression of ERVs in *Pml*^{-/-} mESCs by overexpressing KAP1, an effect that requires active sumoylation, as demonstrated using ML792 (new Fig 4j). Collectively, these results strongly support our model that the repressive effect of PML on ERVs expression is mediated by KAP1 sumoylation.

Reviewer #2 (Remarks to the Author):

PML nuclear bodies represent a distinct subnuclear domain composed of multiple proteins. The cellular function of PML nuclear bodies is still not entirely clear, but circumstantial evidence indicates that these structures are centers for modification with the ubiquitin-related SUMO molecule and stress-induced ubiquitylation of polySUMOylated proteins. Previous work by de The and Lallemand-Breitenbach (Sahin et al., JCB, 2014) proposed that oxidative stress-induced assembly of PML NBs controls SUMOylation of partner proteins. In the current manuscript the authors aimed to identify the PML NB-dependent SUMO targets by combining cell-based and in vivo experiments. Following the identification of potential PML NB-dependent SUMO conjugates the authors concentrate on KAP1/TRIM25 in subsequent functional studies and propose that PML controls the transition of ESCs to the 2-cell like state by enhancing SUMOylation of KAP1 and other key regulators in this pathway. These findings connect published work demonstrating that both SUMO and PML function as important regulators of stem cell pluripotency (Hadjimichael et al., Stem cell reports, 2017; Cossec et al., Cell Stem Cell, 2019).

To my opinion, the manuscript contains a wealth of data that clearly provide several interesting novel aspects, including the identification of candidate PML-dependent SUMO targets and the unravelling of PML's role in orchestrating ESC fate. However, as outlined in detail below some important aspects of the manuscript are not yet fully conclusive. In particular, I feel that the APL model system is not ideal for the identification of PML NB-dependent SUMO substrates by proteomics. Further, the direct link of PML-enhanced SUMOylation to mESC transition is missing. Lastly, the proposed shielding of Ubc9 from oxidation in PML NBs is an appealing concept, but at this stage the data supporting the model are rather preliminary. Generally, the manuscript is not very easy to read and the authors occasionally take argumentative short-cuts to support their claims.

We warmly thank this reviewer for his/her positive comment on the interest and novelty of our study. As detailed below, we have addressed all concerns raised by careful editing and rewriting, together with several novel key experiments. We thank the reviewer for these very detailed comments that helped us to significantly improve the clarity and impact of our manuscript.

1) Based on the experiments in Figure 1 it is concluded that PML NBs mediate basal and stress-induced SUMO conjugation. However, to my opinion, these data are not fully conclusive for several reasons. First, it is concluded that PML controls both basal and stress-induced SUMOylation. The interpretation relies on the comparison of the general SUMOylation pattern observed by immunoblotting in PML^{-/-} mESCs and the corresponding control cells (Figure 1D). There is indeed a huge difference when comparing SUMO2 conjugates in PML^{-/-} and PML^{+/+} mESCs. However, can the authors exclude that the reduced SUMOylation pattern primarily results from the lack of PML itself, which likely represents one of the most abundant SUMO2 target in cells. In line with this possibility PML-SUMO2 conjugates detected in PML^{+/+} cells (Figure 1d, right panel) migrate at exactly the molecular weight of the major SUMO-conjugates. Accordingly, these conjugates are lost in PML^{-/-} cells. The impression that PML controls the bulk of basal SUMOylation may therefore be somewhat misleading, since most targets are simply not detected by immunoblotting.

We thank the reviewer for raising this important and interesting discussion. The wording of the text has been changed to clarify (p5 end of the second paragraph, 6, 7 first paragraph, p13) the fact that in Figure 1, there is circumstantial evidence for a PML effect

on the "bulk of basal sumoylation", while PML dependence upon arsenic stress is clear. The formal demonstration of a role of PML in basal sumoylation of some targets comes later in the manuscript, with the SUMO-proteomics and the unravelling of PML-dependent sumoylation of KAP1 and DPPA2. Yet, it is uneasy to assess how these key targets contribute to the "bulk" of basal sumoylation.

Although PML sumoylation contributes to the overall WB pattern of SUMO2 conjugates, several lines of evidence from Fig. 1 support PML-driven SUMO2 conjugation of other targets. First, it is difficult to state only from similar very high molecular-weight profiles (Fig. 1d) that the increase in total SUMO2/3-modified proteins only reflects increase in PML sumoylation. In fact, some other blots show quite different migration profiles for PML and global SUMO2 targets (Fig. 1g *in vivo*, supplementary Fig. 1h). Second, in His₁₀-SUMO2K/R-expressing mESCs, arsenic has no effect on the amount of SUMO2/3 conjugates, while sumoylated PML is increased (in both amount and molecular weight). This most likely reflects the fact that proteins that would normally be poly-S2-modified cannot form SUMO chains with this SUMO2 mutant, while PML remains multi-sumoylated upon arsenic exposure. Similarly, in His₁₀-S2K/R-expressing APL samples, PML sumoylation sharply increases upon arsenic, while His-purified SUMO2/3 conjugates are unchanged. Moreover, transduction of His₁₀-S2K/R is associated with a dramatic stabilization of SUMO2 conjugates, but not of PML (compare lanes 2&3 to lanes 6&7). Collectively, SUMO2-conjugates cannot not solely reflect sumoylated PML.

The experiments on KAP1 in PML^{-/-} and PML^{+/+} mESCs (Figure 4B) support this view. The data rather show that polySUMOylation (or formation of hybrid SUMO/Ub chains), in particular under stress requires PML.

We are not sure to understand what the reviewer exactly means. There is a reproducible difference in basal KAP1 conjugation by SUMO2/3 between *Pml*^{+/+} and *Pml*^{-/-} mESCs, (bands at 130kDa and above, see Supplementary Fig. 4a and the figures attached for the reviewer).

In addition, PML-dependent KAP1

sumoylation upon arsenic was further supported by our new SUMO proteomics (Fig. 3f and Supplementary Data 4,5). In mESCs, PML also control over basal SUMO2 conjugation of DPPA2 is more obvious than that of KAP1 (Fig. 5d, 4b).

Thus, while we do agree that the PML-dependent stress-induced SUMO2 conjugation is unambiguous, we feel comfortable to state that PML has some effects on basal SUMO2 conjugation.

Second, the data obtained in the APL system are difficult to interpret with respect to the role of PML NBs in basal SUMOylation. The authors argue that in the PML-RAR expressing APL mouse model SUMOylation is impaired due to the loss of PML NB integrity. They further conclude that the restoration of SUMOylation by arsenic treatment is due to the re-assembly of PML NBs. However, the data obtained in the His-SUMO2K/R expressing mice (Figure 1e) are difficult to reconcile with this interpretation. As shown here, robust SUMOylation by His-SUMO2K/R is detected even in the absence of arsenic indicating that SUMOylation occurs very efficiently even when aberrant PML NBs are present in APL cells. Therefore, mono-SUMOylation does not seem to be affected in the APL mice

In line with this interpretation, the MS data suggest that core NB proteins, such as Sp100 exhibit robust SUMOylation in APL cells even without addition of arsenic. The APL model is therefore not the ideal system to explore PML NB-dependent basal SUMOylation.

These comments are interesting and indeed our previous data deserved a more extensive discussion. The reviewer is right, basal sumoylation does occur in APL where PML NBs formation is impaired (rather than abolished), and even for proteins like SP100 known to localize at PML NBs. Our interpretation that enhanced PML NB formation should be accompanied by enhanced target sumoylation, as organelles concentrating/immobilizing enzymes and their substrates. We have further modified the text of the result section, specifying that PML NBs “favor” rather than “drives” “basal in mESCs and stress-induced sumoylation of protein partners, primarily by SUMO2/3 in vivo” (p6, last sentence of the second paragraph, p5 end of the second paragraph, as well as in the discussion, p13)

Critically, we now report analysis of *Pml*^{-/-} APL mice, demonstrating that arsenic no longer promotes SUMO2 conjugation of the targets robustly identified in *Pml*^{+/+} ones (Supplementary Data 1). We have also performed SUMO2 proteomics in arsenic-treated *Pml*^{+/+} versus *Pml*^{-/-} mESCs (see below), with similar conclusions.

2) Data provided in Supplementary Tables 1 +2 do not fully support the conclusion that the proteomics experiment (+/-arsenic) in the APL mice specifically identifies PML NB-dependent SUMO targets. I do acknowledge that a subset of arsenic-induced targets is connected to PML NBs, but the most strongly regulated proteins (e.g. YEATS2, BAZ1B) are - at least to my knowledge- not physically connected to PML or PML NBs. I actually do think that the dataset presented here is very valuable and important to better understand SUMO dynamics in APL cells in response to arsenic treatment.

We thank the reviewer to underline the value of our SUMO proteomic in APL mice. We agree that we could not exclude that the brief arsenic treatment in APL mice modified some SUMO targets independently of PML NB reassembly. We also discuss the possibility that, in this model, PML tethering onto RARA may facilitate the modification of RARA-bound proteins (such as NCORI, Polycomb members ^{6,7,8}) in response to arsenic exposure (p14 first paragraph).

To address this issue, we have now performed two new differential label free quantitative SUMO proteomics experiments. The results fully support our general conclusions.

As mentioned above, one was performed after His₁₀-SUMO2 purification from a novel cohort of *Pml* knockout APL mice. These *Pml*^{-/-} APL mice were treated or not with arsenic. The SUMO proteomics analysis comparing untreated mice with treated ones unraveled PML/RARA as a top hit, as well as RFC1 or Baz1 previously identified in the *Pml*^{+/+} APL cohorts. Critically, most of the other proteins identified as arsenic-modulated

SUMO2/3 targets in *Pml*^{+/+} APL mice, such as KAP1, SP100 or TDG, were not identified in *Pml*^{-/-} APL mice.

A second SUMO proteomics analysis was performed comparing *Pml*^{+/+} and *Pml*^{-/-} His₁₀-SUMO2-expressing mESCs upon arsenic. DPPA2 was among the top differentially sumoylated targets in presence *versus* absence of PML (supporting our focus on this master regulator of 2C/2CL state, Fig. 5). His₁₀-SUMO2 conjugation of KAP1 was also higher in *Pml*^{+/+} compared to *Pml*^{-/-} mESCs. These experiments strongly support the relevance of the results derived from the APL model.

However, at least to my opinion, the experimental setup does not properly reflect the title of the manuscript ("unbiased in vivo exploration of nuclear bodies-enhanced sumoylation...").

We feel that these novel unbiased analyses using SUMO proteomics in mESCs support our conclusions that PML NBs regulate sumoylation of proteins, other than PML itself, and warrant the current title.

3) Following the identification of potential PML NB-dependent SUMO conjugates the authors concentrate on KAP1/TRIM25 in subsequent functional studies. The data support the conclusion that PML favors KAP1 SUMOylation (at least stress-induced polySUMOylation). Previous work has established that KAP1 SUMOylation is involved in transcriptional repression and in particular silences expression of endogenous retrovirus in embryonic stem cells (Yang et al., Cell 2015). Interestingly, the authors show here that PML deficiency also upregulates KAP-1-repressed ERVs possibly connecting PML to this pathway. However, what is missing here is a clear demonstration that PML exerts this effect via enhancing SUMOylation of KAP-1. Chemical or genetic inactivation of SUMOylation is needed to strengthen this point.

We again thank the review for this very relevant remark that overlaps with a comment from reviewer #1. We have now performed the suggested experiments using the ML792 sumoylation inhibitor. While ML792 increased ERV expression in *Pml*^{+/+} cells, it had no significant effects in *Pml*^{-/-} mESCs, in which ERVs are derepressed in PML absence, supporting the notion that this PML effect was mediated by a default of SUMO conjugation (Figure 4i). We also performed a rescue experiment using KAP1-YFP-overexpressing *Pml*^{-/-} mESCs. KAP1-YFP restored ERV repression in *Pml*^{-/-} cells, which was again antagonized by ML792 treatment (Figure 4j), supporting that PML effect on ERVs relies at least in part on sumoylated KAP1 (see text p11). See also the full response to reviewer #1 for other experiments aimed at establishing this point.

4) In the last part of the MS the authors show that PML controls the transition state of ESCs to the 2-cell like state. The transcriptome analysis of PML^{-/-} vs. PML^{+/+} mESCs supports this view and provides a mechanistic basis for this phenotype. Further, it is shown that in PML^{-/-} cells SUMOylation of key regulators of the 2-cell like stage, including DPP2/4 is reduced. Here, the authors need to refer to a recent paper by Theurillat et al. (Cell reports 2020), where the pluripotency factors Dppa2 and Dppa4 were identified as the top-scoring SUMO substrates in ESCs with SUMOylation of Dppa2 and Dppa4 impeding the conversion to 2-cell-embryo-like states.

We thank the reviewer for this reference, which was indeed missing. We have now added this to the text (p12, ref: 53), in complement of the paper by Yan et al. 2019 in Plos Biol.

5) The model that Ubc9 is shielded from oxidation in PML NBs is very speculative at this stage, since the underlying data are rather weak. Generally, this aspect of the manuscript seems quite unrelated to the rest. Further, why was the GRX-roGFP reporter fused to Sp100 and not directly to Ubc9? Moreover, the difference in the bar graph is rather insignificant (Figure 2B).

Since NB biogenesis is controlled by oxidative stress and that the latter inhibits the sumoylation machinery, PML NB-enhanced sumoylation could appear as paradoxical. Measuring the oxidation ratio within the inner core of the bodies could provide one of the clues, although we agree that, at this stage, the evidence presented is circumstantial. Since we could only hypothesize a protective effect on UBC9 function, we followed the reviewer suggestion to tone down the wording when describing these experiments (last paragraph p6) and have removed the sentence from the abstract.

Fusing the reporter to UBC9 could have been another option. Yet, we were concerned that due to reactive cysteine within UBC9, it could have directly contributed to the redox of the GRX-roGFP2, precluding the measurement in the PML NB inner core. Moreover, the efficiency of Sp100 targeting onto NB is much greater than that of UBC9.

Finally, the amplitude of the difference shown is indeed small, but is highly significant, as indicated on the figure.

Where was the PMLE167/- knock-in allele characterized (Figure 2A)? I was unable to find any information in the references that were provided. If no characterization is available it needs to be shown that PMLE167/- indeed affects partner recruitment.

The reviewer is right, this is the first description of these new mice. We now provide a figure characterizing *Pml*^{E167R} knock-in with respect to partner recruitment by showing that, in primary MEFs prepared from these mice, DAXX does not localize with PML E167R bodies (Supplemental Figure 1i).

Additional points:

1. Fig 1a. Loading control for the SUMO2/3 blot is missing. Is it re-probed?

Actually, the samples were the same, migrate twice on a same SDS-PAGE but the membrane was cut into two pieces for incubation with either anti-PML or anti-SUMO2/3 antibodies. We have now modified the figure to add the loading control.

2. Fig 1b. In the chart on Figure 1b colored lines are not labeled, so it is unclear which line correspond to the respective treatments.

We apologize for this technical problem appeared during pdf conversion. It is now corrected.

3. Fig. 1d. What is the significance and rationale of probing His-SUMO1 pull-down membrane with SUMO2/3 and vice-versa? Do the SUMO2/3 blot in the left panel and SUMO1 blot in the right panel indicate abundance of SUMO1+SUMO2/3 conjugated proteins? If so, why is there a difference between the SUMO2/3 blot in the left panel and SUMO1 blot in the right panel?

This is an interesting question that addresses a point not sufficiently detailed in the results. We observed that the fractions of SUMO conjugates dually modified by SUMO1 and SUMO2/3 were not the same when pulling out SUMO1 or SUMO2/3 conjugates. Indeed, the fraction of dually conjugated proteins among the total pool of SUMO1 conjugates or among the total pool of SUMO2 conjugates can be very different. Thus, when SUMO1 or SUMO2 conjugates are purified, their WB analysis will be relative to the total amount of proteins modified by a single paralog. It is thus critically dependent on how “diluted” they are (for example, dually SUMO1/2/3 conjugates might represent 30% of SUMO2/3-modified proteins but 10% of SUMO1 conjugates). Most of the purified SUMO1-conjugated proteins are not or very low SUMO2/3 conjugated in mESCs (previous Fig. 1d). In contrast, among the purified SUMO2/3 conjugates some are also modified by SUMO1. Because botting with anti-SUMO2/3 antibody the His₁₀-SUMO1 conjugates and *vice versa* might be confusing for the readers and our manuscript is not focused on dual conjugation, we have removed these two blots in Figure 1d.

4. Page 5, 3rd line from the top. It is not clear what the authors want to say by “Moreover, in untreated *Pml*^{-/-} mESCs, basal His₁₀-SUMO2 conjugation....by enhanced SUMOylation”? A better explanation would help.

We apologize for this lack of clarity. We observed that SUMO2/3 conjugation decreases in untreated *Pml* KO mESCs compared to WT cells, while that of SUMO1 does not, and is rather higher in these *Pml* KO mESCs, suggesting it might be some compensatory effects between the paralogue. We have rephrased the sentence for clarity (p5).

5. The phrase “Basal levels of global SUMO2 conjugates... catabolism” (Page 5, 6th line from the bottom) is not clear.

We have also reformulated this sentence and we hope that this part of the text is now clearer (p5-6 second, last sentence of the paragraph)

6. Fig. 1f. What is the rationale of SUMO2/3 blot here in the His-SUMO1+HA pull down assay? Why is there a difference (which looks significant) between the 6th and 7th lanes from the left in SUMO2/3 blot, whereas the corresponding HA blot does not show any change?

We performed the blots with anti-SUMO2/3 antibody as a control, notably for multi-sumoylated proteins (like PML) that can display SUMO1 and SUMO2 conjugations on different sites. In all of our experiments, the effect of arsenic was much greater on the conjugation by SUMO2/3, even when SUMO1 conjugates were purified, supporting that upon stress PML NBs primarily favor poly or multiSUMO2/3 conjugation. Pulldown using the His₆ Tag is not as clean as purification using His₁₀ tag, in particular from liver, we thus performed two rounds of purifications (His -pulldown and HA-IP). However, we agree that this is non-essential and could be deleted at the reviewer/editor request.

Lane7, arsenic indeed seems to modestly increase SUMO2 conjugation in *Pml*^{-/-} mouse, although one could argue that there is less protein in lane 6 (as suggested from the RanGap labelling in the HA Western).

7. Fig. 1g. Again, the significance of SUMO1 blot is not clear. Does it represent proteins which are conjugated to both SUMO1 and SUMO2/3? If so, the statement “Preliminary experiments in HA-His₁₀-SUMO3 knock-in mice....SUMO1” should be changed.

Here again, this was essentially a control. There is indeed a significant increase in SUMO1 conjugation of SUMO2/3 adducts. One interpretation could be that in the liver, multi-sumoylated proteins (such as PML) represent a significant fraction of SUMO2/3 conjugates. In any case, we agree that this was unclear and the text was rephrased (p5 line 17).

Reviewer #3 (Remarks to the Author):

In this manuscript, Tessier et al. investigate the role of PML nuclear bodies (NB) in the regulation of embryonic stem cell fate. Using mESCs and mice deficient for PML or PML NB formation (APL mice), combined with expression of His-tagged SUMO variants, the authors demonstrate PML-dependent SUMO conjugation in vitro and in vivo and show that this correlates with stress-induced growth arrest of mESCs. They further show that PML-dependent SUMO conjugation is associated with ubiquitination and degradation of SUMOylated target proteins. Proteomics identification of SUMOylated proteins in APL mice expressing His-tagged SUMO2 revealed known NB components and members of the KAP1 chromatin remodeling complex as SUMO substrates. KAP1 is a master epigenetic regulator known to be involved in stem cell maintenance. The authors validated PML-dependent SUMOylation of KAP1 in mESCs and show that this is associated with PML-dependent repression of transposable elements and the conversion of mESCs into the 2-cell like (2CL) state. The manuscript is well written and properly constructed, and the experimental data is of high quality, but I do have a few major concerns that should be addressed:

We warmly thank this reviewer for this positive appreciation of the data presented and its quality. We have done our best to fully address the major concerns raised in this review.

1) The overall coherence between the different parts and experiments could be improved. Many experiments show interesting associations, but do not show causality, providing rather limited mechanistic insight and often making it difficult to draw strong conclusions. For instance, to show that PML controls the mESC 2CL state via KAP1 SUMOylation, could the authors mutate individual sites in KAP1 or generate a SUMO death mutant of KAP1?

The actual biochemical function of PML bodies was the subject of intense controversies. We establish that partner recruitment in nuclear bodies promotes their SUMO2/3 conjugation, a prelude to the degradation of many of them. In a specific embodiment of this general concept, we discover that PML controls SUMO2 conjugation of essential regulators (DPPA2 and KAP1) in mESCs, opposes their 2C state and modulates their stress responses.

Physiological role of sumoylation is often to fine-tune and facilitate responses to stress through group-modifications of a whole protein complex, rather than black and white impact on a single proteins. It may conceptually not be trivial to find a system where PML absence has clear-cut effects on a single downstream effector to convey its biological impact. We now provide evidence that control of KAP1 sumoylation by PML directly contributes to silencing of some retroviral elements. Indeed, the reviewer is right and the causal link between PML effects on KAP1 sumoylation and KAP1 function was an important issue. To accommodate this point, we have focused on KAP1-mediated ERV regulations (see text p11, Figures 4g-j). This point was also raised by reviewer #1 and we copy our response below.

KAP1 knockout being lethal in mESCs⁵, we could not develop strategies based on inactivation or dead KAP1 mutant Knock In. In addition, to mutate the 6 sites in both *Kap1* alleles of mESCs would be unrealistic. We therefore used *Kap1* siRNA and first confirmed that both *Kap1* and *Ubc9* knockdown (KD) led to ERV de-repression, similar to *Pml* knockout (although of smaller magnitude) (new Fig. 4g). We then explored the effects of *Kap1* KD in a *Pml* KO background, demonstrating that in cells where ERVs

were already derepressed, *Kap1* siRNA did not further enhance expression of endogenous retroviruses (new Fig. 4h). Then as suggested by Reviewer 2, we used the ML792 inhibitor to block sumoylation. Critically, while ML792 enhanced ERV expression in *Pml*^{+/+} cells, it had no significant effects in *Pml* KO mESCs (new Fig. 4i, supplementary Fig. 4d). Finally, we could rescue the repression of ERVs in *Pml*^{-/-} mESCs by overexpressing KAP1, an effect that requires active sumoylation, as demonstrated by its reversal by ML792 (new Fig 4j). Collectively, these results support our model that the repressive effect of PML on ERVs expression is mediated by KAP1 sumoylation. We are grateful for opportunity to improve our study.

2) Regarding the proteomics results in Figure 3:

- Please clearly state in the main text which mouse tissue/organ was analyzed and why this sample type was chosen with regard to mESC biology.

We have now clearly specified these rationales in the text (p7, second paragraph). We choose APL mice for several reasons:

- APL cells massively accumulate in the spleen, allowing the first SUMO proteomic in a cancer *in vivo*.

-APL model -briefly treated or not with arsenic- offers the possibility to study PML NBs-dependent sumoylation rather than simply PML-dependent sumoylation.

-APL model is an asset to analyze sumoylation from *in vivo* and not cell culture since the latter give access to cells already adapted to oxidative stress conditions (20% oxygen in culture, no physiological regulation as in a whole organism), while SUMO2/3 conjugation is known to be exquisitely sensitive to oxidative stress.

Please also note that we have performed two new His₁₀-SUMO2-proteomics experiments to investigate the PML-dependent S2 targets: one using *Pml*^{-/-}APLs treated or not with arsenic and another one comparing arsenic-treated WT mESCs and *Pml* KO mESCs to maximize PML NB biogenesis.

- If I understand correctly, within each cohort 3 comparisons were made, with the first 2 comparisons (un His₁₀-S2 APL versus un Ctrl APL; As His₁₀-S2 APL versus As Ctrl APL) used to select SUMOylated proteins. Selected hits were then used for the 3rd comparison (As His₁₀-S2 APL versus un His₁₀-S2 APL) and plotted in the volcano plots shown in Fig 3b, to see how these SUMOylated proteins have an increase or decrease in SUMOylation upon arsenic acid treatment. This strategy should be clearly described in the main text.

The reviewer has perfectly understood our strategy to perform the differential analyses between arsenic and untreated His₁₀-SUMO2 APL mice, after a first selection of the specific His₁₀-SUMO2 targets. As noticed by the reviewer, we had indeed included several groups of controlled mice (in which the APL cells do not express tagged SUMO2) to allow the first selections of the specific His₁₀-SUMO2 targets. We have also excluded rare proteins with a significant increase between arsenic-treated and untreated Ctrl APL mice, since this does not reflect differences in sumoylation efficiency. For clarity, we have fully described this in the main text and methods.

- I understand that the total proteome was also measured and that the same comparisons were made to calculate protein fold change ratios between the different conditions. Please include an additional Table with the fold change ratios of the shotgun experiments and also add these protein fold change ratios for each SUMO target in Table 1. These data is currently represented

in Supplementary Figure 3, but the heatmap shown there does not allow to lookup fold change ratios for individual proteins. This is particularly important since SUMO2-triggered degradation of proteins is expected over time.

The reviewer is right that the heatmap of the total APL proteome was somehow hard to read. We have now added a table (Supplementary Data 3) with the 173 proteins for which the Anova test under *Perseus* analysis indicated significant changes in total protein abundance between our 3 conditions (untreated, 1h-arsenic, 3h-arsenic). In this table (Supplementary Data 3), we have indicated their fold change and p-value comparing two conditions (Student T-test), similarly to the SUMO proteomic analysis: 1h- or 3h-arsenic-treated *versus* untreated samples.

The total proteomics analysis showed that the abundance of proteins identified in our SUMO-proteomic as arsenic-modulated SUMO2 targets did not change, except Ap3d1. We have thus removed it and we have now listed 88 SUMO2 targets (without duplicats across the cohorts) significantly differentially sumoylated in APL in response to arsenic (Supplementary Data 2).

In Supplementary Data 2, we now provide a comparison between the arsenic-modulated His₁₀-SUMO2 targets from the SUMO2 APL proteomics and arsenic-modulated proteins from the total APL proteomics. We thank the reviewer for his/her useful comment to enhances the clarity of a complicated issue.

- Table 1: please include a significance indication for each protein and each comparison. That will help the reader to understand the selection strategy and is also important to have a complete view of significantly up- or downregulated SUMO targets shown in Fig 3b. At present, it is for instance unclear what are the 19 targets with increased SUMOylation one hour after injection.

We agree with the reviewer that the reading of differential SUMO-proteomics, and their multiple controls, is not so easy. The Source data tables are available on the PRIDE ProteomeXchange deposition website, while Table1 (now Supplementary Data 1) presents data with the proteins selected as His₁₀-SUMO2-specific cleared from contaminants (non-significant proteins based on their indicated p-value and FC, as described in the Methods). Significance (p-value log₁₀) was indicated and we have now added 3 additional columns indicating the selection, for each comparison between His₁₀-SUMO2-APL mice *vs* control APL mice, as well as for the comparison between arsenic-treated *vs* untreated His₁₀-SUMO2 APL. We hope the this will help to understand our selection strategy. We did the same for the new tables (Supplementary Data 4 and 5) and warmly thank the reviewer for raising this point.

- Table 2: for clarity, please also include a sheet that represents the 90 targets mentioned in the main text. Now these targets are distributed over the three sheets, which makes it difficult to follow the behavior of one target over time.

To answer to the reviewer, we have modified Table 2 (now Supplementary Data 2) and the arsenic-specific His₁₀-SUMO2 targets identified within each cohort are now presented on a single sheet. We cannot assemble the data in a single table, because the proteins have been identified from 3 different APL proteomics experiments.

Minor Comments:

- Figure 2A: this panel would fit better in Figure 1 and also the corresponding paragraph in the main text would fit better with the previous section.

We thank the reviewer for this suggestion and both the figure and the corresponding paragraph have been moved as suggested.

- Materials and methods, mass spectrometry data acquisition: please correct sentence at line 9 and 10 of this paragraph

We apologize for this typo error; we have now corrected it.

- Please clarify why the number of missed cleavages is set at an unusually high value of four during the database search.

These setup relies on our past experience in SUMO proteomics. As modification by SUMO or Ubiquitin makes the acceptor lysine non-cleavable by trypsin, increasing the number of allowed missed cleavages to 3-4 compared to the standard of most proteomics analysis by MaxQuant of 2, makes the average score of the peptide-spectrum assignment higher. We used this setup in all our previous SUMO proteomics studies^{9,10,11}.

1. Rowe HM, *et al.* KAP1 controls endogenous retroviruses in embryonic stem cells. *Nature* **463**, 237-240 (2010).
2. Elsasser SJ, Noh KM, Diaz N, Allis CD, Banaszynski LA. Histone H3.3 is required for endogenous retroviral element silencing in embryonic stem cells. *Nature* **522**, 240-244 (2015).
3. Mascle XH, Germain-Desprez D, Huynh P, Estephan P, Aubry M. Sumoylation of the transcriptional intermediary factor 1beta (TIF1beta), the Co-repressor of the KRAB Multifinger proteins, is required for its transcriptional activity and is modulated by the KRAB domain. *The Journal of biological chemistry* **282**, 10190-10202 (2007).
4. Yang BX, *et al.* Systematic identification of factors for provirus silencing in embryonic stem cells. *Cell* **163**, 230-245 (2015).
5. Rowe HM, *et al.* TRIM28 repression of retrotransposon-based enhancers is necessary to preserve transcriptional dynamics in embryonic stem cells. *Genome Res* **23**, 452-461 (2013).
6. Hong SH, David G, Wong CW, Dejean A, Privalsky ML. SMRT corepressor interacts with PLZF and with the PML-retinoic acid receptor alpha (RARalpha) and PLZF-RARalpha oncoproteins associated with acute promyelocytic leukemia. *Proc Natl Acad Sci USA* **94**, 9028-9033 (1997).
7. Minucci S, Nervi C, Lo Coco F, Pelicci PG. Histone deacetylases: a common molecular target for differentiation treatment of acute myeloid leukemias? *Oncogene* **20**, 3110-3115. (2001).
8. Weston AD, Blumberg B, Underhill TM. Active repression by unliganded retinoid receptors in development: less is sometimes more. *The Journal of cell biology* **161**, 223-228 (2003).
9. Hendriks IA, D'Souza RC, Yang B, Verlaan-de Vries M, Mann M, Vertegaal AC. Uncovering global SUMOylation signaling networks in a site-specific manner. *Nature structural & molecular biology* **21**, 927-936 (2014).
10. Hendriks IA, Lyon D, Young C, Jensen LJ, Vertegaal AC, Nielsen ML. Site-specific mapping of the human SUMO proteome reveals co-modification with phosphorylation. *Nature structural & molecular biology*, (2017).

11. Gonzalez-Prieto R, *et al.* Global non-covalent SUMO interaction networks reveal SUMO-dependent stabilization of the non-homologous end joining complex. *Cell reports* **34**, 108691 (2021).

REVIEWERS' COMMENTS

Reviewer #1 (Remarks to the Author):

Tessier et al, 2022

The revised version of the manuscript is substantially improved. The authors employ a strategy of chemical inhibition to examine the involvement of KAP1 sumoylation in the changes that occur in mESC in the absence PML. They also identify sumo 2 targets in Arsenate treated control and PML -/- mESC which is a much better model than APL mice. In the case of endogenous retroviruses they show that ML792 enhanced ERV expression in Pml+/+ but not PML-/- cells. Overexpression of KAP1 was able to restore the repression of ERVs in Pml-/- mESCs but not in the presence of ML792 clearly demonstrating the role of PML –dependent KAP1 sumoylation in the derepression of retroelements. However in the case of 2C state transition, the authors limit their analysis to the detection of Dppa2 as a PML dependent sumo 2 target. It would be advisable that the authors employ the sumo inhibitor ML792 to assess the role of KAP1 sumoylation in the case of 2C transition.

Reviewer #2 (Remarks to the Author):

In their revised manuscript the authors made considerable efforts to address the concerns I raised on the initial version. Novel data and a more thorough discussion of some issues have considerably strengthened the manuscript. I particularly appreciate that point my #3, (direct link of PML-driven sumoylation of KAP1 in restraining ESC to 2C conversion) has been clarified. Regardless of that, I still do have concerns on some other issues. This particularly concerns the role of PML NBs in favouring local ROS protection (my point #5). To my opinion the data is still not fully conclusive and the interpretation remains speculative. Given that - at least to my opinion - this aspect does not add much to the story, I suggest to remove it. Generally, the manuscript is still not easy to read and should go through a thorough redactional "polishing".

Reviewer #3 (Remarks to the Author):

I appreciate the efforts from the authors to address my comments in depth. The proteomics data is now more clearly presented, both in the figures and Supplementary Data files. This has strongly improved the clarity of the manuscript which can now be accepted for publication.

Point by point response to reviewers' second comments

Reviewer #1 (Remarks to the Author):

The revised version of the manuscript is substantially improved. The authors employ a strategy of chemical inhibition to examine the involvement of KAP1 sumoylation in the changes that occur in mESC in the absence PML. They also identify sumo 2 targets in Arsenate treated control and PML $-/-$ mESC which is a much better model than APL mice. In the case of endogenous retroviruses they show that ML792 enhanced ERV expression in Pml $+/+$ but not PML $-/-$ cells. Overexpression of KAP1 was able to restore the repression of ERVs in Pml $-/-$ mESCs but not in the presence of ML792 clearly demonstrating the role of PML $-$ dependent KAP1 sumoylation in the derepression of retroelements.

We warmly thank this reviewer to underscore the input of the new data to the revised version.

However, in the case of 2C state transition, the authors limit their analysis to the detection of Dppa2 as a PML dependent sumo 2 target. It would be advisable that the authors employ the sumo inhibitor ML792 to assess the role of KAP1 sumoylation in the case of 2C transition.

Concerning the mechanism by which PML may restrict 2CL transition, in addition to DPPA2 sumoylation in Pml $^{+/+}$ mESCs, we provided mESC-SUMO2 proteomics analysis in the revised manuscript which identified both DPPA2 and DPPA4 as top PML-dependent SUMO2 targets. Yet, we do agree with the reviewer that investigating the role of SUMO on 2CL transition using ML792 in Pml $^{+/+}$ or Pml $^{-/-}$ mESCs is of interest. We now report the expression levels of *Dux* and *Mael*, two DPPA2-target genes, upon sumoylation inhibition in Pml $^{+/+}$ and in Pml $^{-/-}$ mESCs (Supplementary figure 4i). Remarkably, ML792 increases their transcription levels in Pml $^{+/+}$ cells, but not Pml $^{-/-}$ ones, in which DPPA2 sumoylation is defective.

Reviewer #2 (Remarks to the Author):

In their revised manuscript the authors made considerable efforts to address the concerns I raised on the initial version. Novel data and a more thorough discussion of some issues have considerably strengthened the manuscript. I particularly appreciate that point my #3, (direct link of PML-driven sumoylation of KAP1 in restraining ESC to 2C conversion) has been clarified.

We thank the reviewer for highlighting our efforts to address all the points raised previously and the strength of our new data.

Regardless of that, I still do have concerns on some other issues. This particularly concerns the role of PML NBs in favouring local ROS protection (my point #5). To my opinion the data is still not fully conclusive and the interpretation remains speculative. Given that - at least to my opinion - this aspect does not add much to the story, I suggest to remove it. "

We agree that the redox status of PML NBs shielding UBC9 from local ROS remains somehow speculative and, furthermore, may not be restricted to the SUMO-E2. Indeed,

it may apply to many enzymes with cysteine-containing active sites. We thus modified the text in that sense and tuned down our conclusions.

We think that sharing this observation with the community may nevertheless be useful to reconcile two apparently paradoxical observations: increase in sumoylation or ubiquitination upon oxidative stress and inactivation of E1 and E2 SUMO and ubiquitin enzymes by ROS *in cellulo* or *in vitro*. We could accept to delete this part at the editors' request.

Generally, the manuscript is still not easy to read and should go through a thorough redactional "polishing"

We have extensively polished the main text and hope that it now fulfils the criteria of this reviewer.

Reviewer #3 (Remarks to the Author):

I appreciate the efforts from the authors to address my comments in depth. The proteomics data is now more clearly presented, both in the figures and Supplementary Data files. This has strongly improved the clarity of the manuscript which can now be accepted for publication.

We warmly thank this reviewer for this positive appreciation of our data presentation and quality of the manuscript.